# Scaling up Continuous-Time Markov Chains Helps Resolve Underspecification

**Alkis Gotovos**
MIT
alkisg@mit.edu

**Rebekka Burkholz**
Harvard University
rburkholz@hsph.harvard.edu

**John Quackenbush**
Harvard University
johnq@hsph.harvard.edu

**Stefanie Jegelka**
MIT
stefje@mit.edu

## Abstract

Modeling the time evolution of discrete sets of items (e.g., genetic mutations) is a fundamental problem in many biomedical applications. We approach this problem through the lens of continuous-time Markov chains, and show that the resulting learning task is generally underspecified in the usual setting of cross-sectional data. We explore a perhaps surprising remedy: including a number of additional independent items can help determine time order, and hence resolve underspecification. This is in sharp contrast to the common practice of limiting the analysis to a small subset of relevant items, which is followed largely due to poor scaling of existing methods. To put our theoretical insight into practice, we develop an approximate likelihood maximization method for learning continuous-time Markov chains, which can scale to hundreds of items and is orders of magnitude faster than previous methods. We demonstrate the effectiveness of our approach on synthetic and real cancer data.

## 1 Introduction

Modeling the time evolution of physical processes with discrete states is an important machine learning problem that spans a wide range of application domains, including molecular dynamics (Crommelin & Vanden-Eijnden, 2006), phylogenetics (Suchard et al., 2001), and computational medicine (Liu et al., 2015). Continuous-time markov chains have found success in such modeling tasks, supported by extensive research into learning and inference in such models (Opper & Sanguinetti, 2008; Perkins, 2009; Archambeau & Opper, 2011; Rao & Teh, 2012).

While the majority of previous work has focused on modeling a relatively small number of discrete states, many interesting applications involve the interaction between sets of items, which results in exponentially large state spaces. For example, when modeling the time progression of cancer, we want to consider a number of genetic alterations (e.g., mutations, copy number variations, etc.) that exhibit complex time-related dependencies (Beerenwinkel et al., 2014). To model how sets of $n$ such alterations evolve over time, we need to consider a state space of size $2^n$. What makes the problem even more challenging is the fact that the available data sets are typically cross-sectional; that is, they consist of (unordered) sets of items observed at unknown time points without any further information about the history of the underlying process.

In this paper, we focus on the problem of learning a particular parametric family of continuous-time Markov chains from such cross-sectional data, and show that the resulting problem is in general underspecified. The issue of underspecification is many-faceted and has been shown to permeate a

35th Conference on Neural Information Processing Systems (NeurIPS 2021).

large number of machine learning systems, often leading to poor generalization, lack of robustness, and spurious relationships when interpreting the resulting models (D'Amour et al., 2020). In our setting, we explore a perhaps surprising remedy: including a number of additional (approximately) independent items can help determine the time order of process events, and hence resolve underspecification. We theoretically show that these extra items act as a "background clock", since counting the number of their occurrences in a data sample can help us estimate the time at which that sample was observed. In sharp contrast to common practice, which limits the analysis to a small subset of relevant, highly-interacting items, our insight suggests that scaling up the learning procedure can be crucial for the robustness of the inferred models. We thus show that items deemed a priori unimportant to the application at hand may in fact be particularly valuable for recovering the time properties of the underlying physical process.

Existing learning approaches are, unfortunately, not well-suited for practically applying this insight; for example, the state-of-the-art method by Schill et al. (2019) for learning continuous-time Markov chains to model cancer progression, scales exponentially in the number of items considered, thus limiting the analysis to around 20 items. To alleviate this issue, we propose an approximate likelihood maximization method that relies on a fast gradient approximation. On problems of 20 items our approach runs almost 1000 times faster that the state of the art, while it can also scale to problems involving hundreds of items. In experiments on real cancer data, we demonstrate how some previous results may have been artifacts of underspecification, and how scaling up the analysis can result in more robust models of cancer progression.

## 2   Problem setup

Given a ground set $V = \{1, \ldots, n\}$, we consider a continuous-time Markov chain (CTMC) $\{X(t)\}_{t \geq 0}$ on state space $2^V$ (Grimmett & Stirzaker, 2001). Thus, the states of the Markov chain are subsets of $V$, and can equivalently be identified as binary vectors of size $n$. CTMCs are commonly represented by a generator matrix $Q \in \mathbb{R}^{2^n \times 2^n}$. Since the rows and columns of this matrix correspond to states in $2^V$, we will use $q_{S \to R}$ to denote the entry of $Q$ at row indexed by $S$ and column indexed by $R$. For $S \neq R$, the entry $q_{S \to R}$ represents the infinitesimal rate of transitioning from state $S$ to state $R$, that is,

$$q_{S \to R} = \lim_{\delta t \to 0^+} \frac{\mathbb{P}(X_{t+\delta t} = R \mid X_t = S)}{\delta t}, \;\; \text{for } S \neq R,$$

with $q_{S \to R} \geq 0$. The quantity $\tilde{q}_S := -q_{S \to S}$ represents the total infinitesimal rate of leaving state $S$, with $\tilde{q}_S \geq 0$. Since the total rate of leaving $S$ is equal to the sum of the rates of transitioning to any other state $R$, we have $\tilde{q}_S = \sum_{R \in 2^V} q_{S \to R}$, for all $S \in 2^V$.

We are interested in modeling physical processes that induce an increasing accumulation of items over time, for example, the accumulation of genetic alterations when modeling cancer progression. To this end, we impose the following two constraints on the CTMC. First, we assume that the chain always starts from the empty state, that is, $X(0) = \emptyset$. Second, we only allow transitions that add a single item to the current set, that is, for $S \neq R$, $q_{S \to R} \neq 0$ only if $R = S \cup \{i\}$ for some $i \in V$. Since items are never removed, this implies that $\lim_{t \to \infty} X_t = V$.

In this paper, we focus on learning the CTMC from a given data set. Since learning directly the exponentially sized generator $Q$ is a hopeless endeavor, we impose a particular parametric form on this matrix, recently introduced by Schill et al. (2019). We define a parameter matrix $\Theta \in \mathbb{R}^{n \times n}$ (also denoted by $\theta$ as a vector), and parameterize the off-diagonal entries of the generator matrix as

$$q_{S \to S \cup \{j\}}(\boldsymbol{\theta}) = \exp\left(\theta_{jj} + \sum_{i \in S} \theta_{ij}\right),$$

for all $S \subseteq V$, and $j \in V \setminus S$. The off-diagonal elements of $\Theta$ encode positive (attractive) or negative (repulsive) pairwise interactions between items in $V$. The presence of item $i$ increases ($\theta_{ij} > 0$) or decreases ($\theta_{ij} < 0$) the rate of adding item $j$ by a multiplicative factor of $w_{ij} := e^{\theta_{ij}}$. The diagonal elements $\theta_{jj}$ encode the intrinsic rate of adding element $j$ when ignoring all other interactions.

Obtaining a sample from a CTMC under all aforementioned constraints can be done using the following sequential procedure. At each step, given the current state $S$, we draw a "holding time" $h$

**Algorithm 1:** Sampling a set from the marginal CTMC

---

**Input :** Parameters $\boldsymbol{\theta}$
Draw $t_{\text{obs}} \sim \text{Exp}(1)$
$t \leftarrow 0, \ S \leftarrow \emptyset$
**while** $t < t_{\text{obs}}$ **do**
    Draw $h \sim \text{Exp}(\tilde{q}_S(\boldsymbol{\theta}))$
    Compute $p_i = q_{S \to S \cup \{i\}}(\boldsymbol{\theta})/\tilde{q}_S(\boldsymbol{\theta})$, for $i \in V \setminus S$
    Draw $x \sim \text{Cat}(V \setminus S, (p_i)_{i \in V \setminus S})$
    $t \leftarrow t + h, \ S \leftarrow S \cup \{x\}$
**return** $S$

---

from an exponential distribution with parameter $\tilde{q}_S$. This represents how much time will pass until we add the next item. Then, we draw the next item $x \in V \setminus S$ according to a categorical distribution with probabilities $\propto q_{S \to S \cup \{x\}}$. The result is a sequence of items $(\sigma_1, \ldots, \sigma_n)$, $\sigma_i \in V$, together with a sequence of holding times $(h_1, \ldots, h_n)$, $h_i \in \mathbb{R}$.

Ideally, our data set would consist of a number of such item sequences and holding times; we would then proceed to learn the parameters $\boldsymbol{\theta}$ by maximizing the likelihood of our CTMC model. Unfortunately, such detailed data is typically not available in practice. In particular, it is rather common in biomedical applications to only have access to cross-sectional data, i.e., to only observe the current state (in the form of an unordered subset of $V$) at a particular point in time without knowing how the process reached that state. Furthermore, it is often the case that the time point of each observation can only be roughly estimated or, worse, is completely unknown.

Similarly to previous work, we assume that we are given a data set $\mathcal{D} = \{S^{(1)}, \ldots, S^{(N)}\}, S^{(i)} \subseteq V$, with each $S^{(i)}$ representing a state observed at a potentially different unknown time $T_{\text{obs}}$. More specifically, we assume that $T_{\text{obs}}$ is a random variable following an exponential distribution with density $p(t) = e^{-t}$ (Gerstung et al., 2009; Schill et al., 2019), and investigate the problem of maximizing the marginal likelihood,

$$p(\mathcal{D}; \boldsymbol{\theta}) = \prod_{i=1}^{N} p(S^{(i)}; \boldsymbol{\theta}) = \prod_{i=1}^{N} \int_0^\infty p(S^{(i)} \,|\, t; \boldsymbol{\theta}) p(t) dt. \tag{1}$$

Algorithm 1 shows how to sample a set from $p(S; \boldsymbol{\theta})$. The difference from our previous description of sampling from a CTMC lies in the fact that the procedure is only run until we reach the observation time $t_{\text{obs}}$, and the result is an unordered set without any information about the holding times.

## 3 The value of "unimportant" items

Our ultimate goal is to retrieve the ordered interaction structure present in the data by learning the parameter vector $\boldsymbol{\theta}$. Before attempting to maximize the marginal likelihood of eq. (1), though, it is natural to first ask whether the limited information available in the data is sufficient to infer such ordered interactions in the first place. We start with a simple example to showcase this issue.

### 3.1 Warmup example

Consider a ground set $V = \{1, 2\}$ and true model parameters $\theta_{11}^* = \theta_{21}^* = 0$, and $\theta_{12}^* = -\theta_{22}^* = \alpha$, for some $\alpha > 0$. When $\alpha$ is sufficiently large, the probability of item 2 occurring before item 1 goes to zero, therefore the model encodes the fact that item 1 is a prerequisite (and therefore, always appears before) item 2. As a consequence, we will also have $p(\{2\}; \boldsymbol{\theta}^*) \approx 0$ for the resulting marginal distribution of sets. One may intuitively think that this could be enough information to infer the true time order; the following proposition shows that this is not the case.

**Proposition 1.** *There is a one-dimensional family of parameters* $\boldsymbol{\theta} = \boldsymbol{\theta}(s)$, *and* $s_1, s_2 \in \mathbb{R}$, *such that* $p(\cdot; \boldsymbol{\theta}(s)) = p(\cdot; \boldsymbol{\theta}^*)$, *for all* $s_1 < s < s_2$.

The proof follows by simple algebra, and can be found in Appendix A together with an illustration of the parameter family. Interestingly, for $s \to s_1$, the model $\boldsymbol{\theta}(s)$ encodes the fact that, when we

observe both items, item 2 always appears before item 1, which is the opposite of what is encoded in our true model.

## 3.2 Independent items as a background clock

While the above example paints a pessimistic picture for inferring the correct time order, we show that this is still possible given some additional side information. Suppose that we are given another ground set $V_+$ whose items have no interaction with the items in $V$, that is, the parameter matrix $\Theta_{\text{full}}$ of the resulting model over $V_{\text{full}} = V \cup V_+$ has a block diagonal structure with blocks $\Theta \in \mathbb{R}^{|V| \times |V|}$ and $\Theta_+ \in \mathbb{R}^{|V_+| \times |V_+|}$. Then, it is easy to see that the distributions over $V$ and $V_+$ are conditionally independent given time $t$. That is, for any $S \subseteq V$, $S_+ \subseteq V_+$, we can write $p(S \cup S_+ \,|\, t; \boldsymbol{\theta}_{\text{full}}) = p(S \,|\, t; \boldsymbol{\theta}) \, p(S_+ \,|\, t; \boldsymbol{\theta}_+)$. Note that the same statement does not hold when considering the marginal distributions, i.e., $p(S \cup S_+; \boldsymbol{\theta}_{\text{full}}) \neq p(S; \boldsymbol{\theta}) \, p(S_+; \boldsymbol{\theta}_+)$. Using this conditional independence property, we can rewrite the marginal probability of $S \cup S_+$ as follows:

$$
\begin{aligned}
p(S \cup S_+; \boldsymbol{\theta}_{\text{full}}) &= \int_0^\infty p(S \cup S_+ \,|\, t; \boldsymbol{\theta}_{\text{full}}) p(t) dt \\
&= \int_0^\infty p(S \,|\, t; \boldsymbol{\theta}) \, p(S_+ \,|\, t; \boldsymbol{\theta}_+) p(t) dt && \text{(by cond. ind.)} \\
&= \int_0^\infty p(S \,|\, t; \boldsymbol{\theta}) \, p(t \,|\, S_+; \boldsymbol{\theta}_+) p(S_+; \boldsymbol{\theta}_+) dt && \text{(by Bayes' rule)} \\
\Rightarrow \quad p(S \,|\, S_+; \boldsymbol{\theta}_{\text{full}}) &= \int_0^\infty p(S \,|\, t; \boldsymbol{\theta}) \, p(t \,|\, S_+; \boldsymbol{\theta}_+) dt && \text{(dividing with } p(S_+; \boldsymbol{\theta}_+) \text{).} \quad (2)
\end{aligned}
$$

Comparing equations (1) and (2), we see that the information about $S$ gained by observing $S_+$ can be explained via a posterior observation time distribution $p(t \,|\, S_+; \boldsymbol{\theta}_+)$ that refines the prior $p(t)$ by taking into account $S_+$.

To gain further insight into this time posterior, we analyze in more detail a simplified setup, in which $V_+$ consists of $m$ identically distributed independent items parameterized by $\theta_+$, that is, $\Theta_+ = \theta_+ \boldsymbol{I}_m$. In this case, $|S_+| \,|\, t$ follows a binomial distribution over $m$ variables with success parameter $1 - e^{-w_+ t}$, where $w_+ := e^{\theta_+}$. Intuitively, we expect these independent items to act as a "background clock": counting the number of items $|S_+|$ should give us an indication about the observation time $t$. Moreover, as we increase the number of items $m$ we should be able to get increasingly accurate estimates of the observation time. The following theorem formalizes these ideas.

**Theorem 1.** *Let $S_+ \subseteq V$ be randomly drawn according to the CTMC with parameter matrix $\Theta_+$, and let $t^*$ be the true observation time of $S_+$. Then, for any $\delta \in (0, 1)$, there exists an $m_0$, such that for all $m \geq m_0$, the mean and variance of the posterior time distribution $p(t \,|\, S_+; \boldsymbol{\theta})$ can be bounded as follows with probability at least $1 - \delta$,*

$$
|M_{post} - t^*| \leq C_1(\theta_+, t^*) \sqrt{\frac{\log m}{m}} + \mathcal{O}\left(\frac{\log m}{m}\right)
$$

$$
V_{post} \leq C_2(\theta_+, t^*) \frac{1}{m} + \mathcal{O}\left(\frac{1}{m^2}\right).
$$

The quantities $C_1$ and $C_2$ are constant w.r.t. $m$ and encode how suitable a particular rate (quantified through $\theta_+$) of an item is for estimating the specific observation time $t^*$. Intuitively, larger rates are better suited for estimating smaller times and vice versa. We further illustrate this effect in Appendix B together with the detailed proof of the theorem. Note that for $m \to \infty$ we get $p(t \,|\, S_+; \boldsymbol{\theta}_+) \to \delta(t - t^*)$ in distribution, and therefore $p(S \,|\, S_+; \boldsymbol{\theta}_{\text{full}}) \to p(S \,|\, t^*; \boldsymbol{\theta})$. This means that observing an arbitrarily large amount of independent items is equivalent to knowing the true observation time. In Appendix C we show that a similar asymptotic convergence behavior can be observed even when the items in $V_+$ are not identically distributed or exhibit some dependencies.

## 3.3 Discussion

To illustrate how this result applies to our two-item example discussed before, we define $\Theta_+ = \boldsymbol{I}_m[\theta_+^{(1)}, \ldots, \theta_+^{(m)}]^T$, where each $\theta_+^{(i)}$ is drawn uniformly in $[-4, -2]$. This corresponds to individual

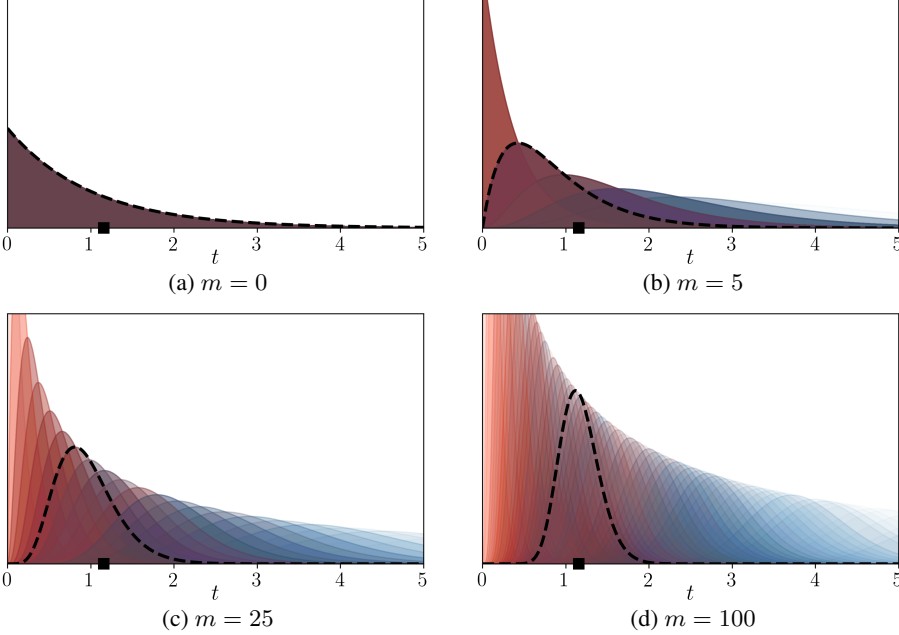

Figure 1: Posterior time densities for $N = 500$ samples. Posteriors of samples with $S = \{1\}$ are colored red, while those of samples with $S = \{1, 2\}$ are colored blue. The dashed line highlights the time posterior of a specific sample, while the black square denotes its true observation time.

frequencies of items in $S_+$ that roughly range from 0.01 to 0.1. We then draw $N = 500$ samples from the CTMC defined by $\boldsymbol{\theta}_{\text{full}}$. Figure 1 shows the posterior time density corresponding to each sample with $S = \{1\}$ (red) and $S = \{1, 2\}$ (blue), for different numbers $m$ of independent items. To avoid clutter, we do not show the densities for samples with $S = \emptyset$. As expected, when $m = 0$ all samples follow the prior time density, and we have no way to distinguish their time order. However, as we keep increasing $m$ we notice that the two colors keep separating from each other in time, and it becomes clear that the red posteriors tend to concentrate around earlier observation times than the blue ones. This is a clear indication that item 1 comes before item 2, and the additional time information obtained by the posterior time estimates should help pinpoint the true model within the family discussed in Proposition 1.

Due to computational considerations, it is common practice to constrain the ground set to a small subset of items that interact strongly and are deemed interesting to the application at hand; for example, genetic alterations that are known to have a role in cancer development (Raphael & Vandin, 2015; Schill et al., 2019). Our results in this section challenge this practice. We show that items believed to be independent of the "interesting items", and thus considered unimportant to the analysis, can in fact be particularly valuable in estimating the observation time of data samples and recovering the time order properties of the underlying physical process.

For the remainder of this paper, we will not assume any prior knowledge about the block-diagonal structure of $\boldsymbol{\Theta}_{\text{full}}$ or the values of the parameters $\boldsymbol{\theta}_+$. Furthermore, we will make no distinction between "items of interest" and independent items. If there is indeed a block-diagonal parameter structure in the data, the addition of a regularization term that promotes sparsity can help recover this structure in the presence of noise. For this reason, similarly to what was used by Schill et al. (2019), the final objective we maximize in practice is the $L_1$-regularized marginal log-likelihood,

$$F(\mathcal{D}; \boldsymbol{\theta}) = \frac{1}{N} \sum_{d=1}^{N} \log p(S^{(d)}; \boldsymbol{\theta}) - \lambda \sum_{i \neq j} |\theta_{ij}|. \tag{3}$$

# 4 Efficient approximate likelihood maximization

To put the previously discussed theoretical insight into practice, we need to be able to efficiently maximize the above objective for ground sets $V$ containing potentially hundreds of items. When using a first-order method for optimization, the bulk of the required computation is devoted to obtaining the gradient of the marginal log-likelihood $\log p(S; \boldsymbol{\theta})$ with respect to the parameters $\boldsymbol{\theta}$. Computing the exact gradient requires $\mathcal{O}(2^n)$ computation; we propose here a method to compute a gradient approximation in a much more efficient manner. We start by deriving expressions for the likelihood in some simplified setups, and then use these results as building blocks for our setup of interest.

## 4.1 Full sequences

First, we consider a setup in which we observe the arrival order of the items in the CTMC, which corresponds to building up an ordered sequence $\sigma$ instead of a set $S$ in Algorithm 1. Furthermore, we assume that $t_{\text{obs}} = +\infty$, which means that the returned sequence will be a permutation of the ground set $V$. By definition of the CTMC, the probability of observing a sequence $\sigma = (\sigma_1, \ldots, \sigma_n) \in \mathcal{S}_V$, where $\mathcal{S}_V$ denotes the permutation group over $V$, can be written as

$$p(\sigma; \boldsymbol{\theta}) = \prod_{i=1}^{n} \mathbb{P}\left(X_i = \sigma_{[i]} \mid X_{i-1} = \sigma_{[i-1]}\right) = \prod_{i=1}^{n} \frac{q_{\sigma_{[i-1]} \to \sigma_{[i]}}(\boldsymbol{\theta})}{\tilde{q}_{\sigma_{[i-1]}}(\boldsymbol{\theta})}. \tag{4}$$

We denote subsequences using the bracket notation, $\sigma_{[i]} := (\sigma_1, \ldots, \sigma_i)$, with $\sigma_{[0]} = ()$. Also, with a slight abuse of notation we directly use sequences in some places that require sets; therefore, the notation $\tilde{q}_\sigma$ is to be understood as $\tilde{q}_{\text{set}(\sigma)}$, where $\text{set}(\sigma) := \{\sigma_i \mid i \in \{1, \ldots, |\sigma|\}\}$.

## 4.2 Partial sequences given time

As a next step, instead of $t_{\text{obs}} = +\infty$, we assume that we are given the observation time $t_{\text{obs}}$, and would like to derive the probability of observing a partial sequence $\sigma = (\sigma_1, \ldots, \sigma_k)$, with $k \leq n$, at time $t_{\text{obs}}$. We can express this as the probability of the intersection of two events in the space of outcomes of the continuous-time Markov chain. Event $\mathcal{A}$ identifies the outcomes in which the first $k$ ordered elements agree with our partial sequence $\sigma$. Event $\mathcal{B}$ identifies the outcomes for which the observation time $t_{\text{obs}}$ falls between the addition of the $k$-th and $k+1$-th elements; that is, $T_k < t_{\text{obs}} < T_{k+1}$. We can then express the desired probability as $p(\sigma \mid t_{\text{obs}}; \boldsymbol{\theta}) = \mathbb{P}(\mathcal{A}) \mathbb{P}(\mathcal{B} \mid \mathcal{A})$.

The probability of event $\mathcal{A}$ can be directly written using (4), except that, in this case the product is taken from $i = 1$ to $k$, instead of $n$. Regarding event $\mathcal{B}$, first note that the sequence of events $\left(\{T_i < t_{\text{obs}} \mid \mathcal{A}\}\right)_{i=1}^{n}$ is decreasing. As a result, we can write

$$\mathbb{P}(\mathcal{B} \mid \mathcal{A}) = \mathbb{P}(T_k < t_{\text{obs}} < T_{k+1} \mid \mathcal{A}) = \mathbb{P}(T_k < t_{\text{obs}} \mid \mathcal{A}) - \mathbb{P}(T_{k+1} < t_{\text{obs}} \mid \mathcal{A}). \tag{5}$$

By definition of the CTMC, each holding time $H_i$ is an exponentially distributed random variable whose parameter depends on the $i - 1$ items that have been added up to that point. We can also define the jump time $T_k := \sum_{i=1}^{k} H_i$, which represents the time at which the $k$-th change of state occurs. When we condition on knowing the first $k$ items of the sequence, i.e., event $\mathcal{A}$, the jump times $T_k$ and $T_{k+1}$ are distributed as sums of $k$ and $k+1$ independent exponential random variables respectively. The CDF of the sum of $r$ independent exponential variables with rates $\lambda_1, \ldots, \lambda_r$, also known as the hypoexponential distribution (Bibinger, 2013), is given by

$$F_{\text{HEXP}}(y; \lambda_1, \ldots, \lambda_r) = \left(\prod_{i=1}^{r} \lambda_i\right) \sum_{i=1}^{r} \frac{1 - e^{-\lambda_i y}}{\lambda_i \prod_{j \neq i}(\lambda_j - \lambda_i)}.$$

In our case, the rates $\lambda_i$ of each exponential distribution are $\tilde{q}_{[\sigma_i]}(\boldsymbol{\theta})$, thus we can compute the terms of (5) as $\mathbb{P}(\mathcal{B} \mid \mathcal{A}) = F_{\text{HEXP}}(t_{\text{obs}}; \tilde{q}_{[\sigma_1]}, \ldots, \tilde{q}_{[\sigma_k]}) - F_{\text{HEXP}}(t_{\text{obs}}; \tilde{q}_{[\sigma_1]}, \ldots, \tilde{q}_{[\sigma_{k+1}]})$.

## 4.3 Marginal partial sequences

Next we consider the probability of observing a partial sequence $\sigma = (\sigma_1, \ldots, \sigma_k)$, with $k \leq n$, without knowledge of the observation time $t_{\text{obs}}$. We define events $\mathcal{A}$ and $\mathcal{B}$ as before, except that in

this case the observation time $T_{\text{obs}}$ is a random variable, and event $\mathcal{B}$ is defined by $T_k < T_{\text{obs}} < T_{k+1}$. In particular, we assume that $T_{\text{obs}}$ is an exponential random variable with rate $\lambda_{\text{obs}} = 1$. Rather than naively integrating over time the probability derived before, we can obtain the following greatly simplifed expression by making use of the memoryless property of exponential random variables (Bertsekas & Tsitsiklis, 2008) when computing $\mathbb{P}(\mathcal{B} \mid \mathcal{A})$.

**Proposition 2.** *The marginal probability of a partial sequence $\sigma = (\sigma_1, \ldots, \sigma_k)$ can be written as*

$$p(\sigma; \boldsymbol{\theta}) = \left( \prod_{i=1}^{k} \frac{q_{\sigma_{[i-1]} \to \sigma_{[i]}}(\boldsymbol{\theta})}{1 + \tilde{q}_{\sigma_{[i-1]}}(\boldsymbol{\theta})} \right) \frac{1}{1 + \tilde{q}_{\sigma_{[k]}}(\boldsymbol{\theta})}. \tag{6}$$

The proof can be found in Appendix D, but the form of this expression allows for an intuitive interpretation. We can modify the original continuous-time Markov chain by adding an extra dummy state $X_{\text{obs}}$, such that for all $X \subseteq V$, $q_{X \to X_{\text{obs}}}(\boldsymbol{\theta}) = 1$, and $q_{X_{\text{obs}} \to X}(\boldsymbol{\theta}) = 0$. The first property implies that there is a fixed rate 1 to transition from any state to $X_{\text{obs}}$, while the second property implies that $X_{\text{obs}}$ is a terminal state. It is easy to see then, that eq. (6) expresses exactly the probability of observing a "full" sequence $\sigma$ in this modified chain. Note that a "full" sequence in this chain is no longer a permutation of $V$, but rather any sequence that ends with $X_{\text{obs}}$.

### 4.4 Marginal sets

Finally, we consider the probability of observing a set $S \subseteq V$, without knowledge of the order in which the individual items arrived, and still under the assumption that the observation time $T_{\text{obs}}$ is an exponential random variable with rate $\lambda_{\text{obs}} = 1$. Making use of the result in the previous section, we can compute this probability by summing over all partial sequences that are permutations of $S$, that is, $p(S; \boldsymbol{\theta}) = \sum_{\sigma \in \mathcal{S}_S} p(\sigma; \boldsymbol{\theta})$. However, this computation is infeasible for all but very small set sizes, since it requires summing over $|S|!$ terms. To alleviate this problem, we propose a method to approximate the gradient of $\log p(S; \boldsymbol{\theta})$. This gradient can be written as

$$\nabla_{\boldsymbol{\theta}} \log p(S; \boldsymbol{\theta}) = \frac{1}{p(S; \boldsymbol{\theta})} \nabla_{\boldsymbol{\theta}} p(S; \boldsymbol{\theta}) = \frac{1}{p(S; \boldsymbol{\theta})} \sum_{\sigma \in \mathcal{S}_S} \nabla_{\boldsymbol{\theta}} p(\sigma; \boldsymbol{\theta}) = \sum_{\sigma \in \mathcal{S}_S} \frac{p(\sigma; \boldsymbol{\theta})}{p(S; \boldsymbol{\theta})} \nabla_{\boldsymbol{\theta}} \log p(\sigma; \boldsymbol{\theta}).$$

In the last expression, we observe that the gradient corresponding to each permutation $\sigma$ is weighed by the probability of observing that permutation of the given set $S$. This suggests a stochastic approximation of the desired gradient by first sampling $m$ permutations $\sigma^{(1)}, \ldots, \sigma^{(M)}$ according to $p(\cdot \mid S; \boldsymbol{\theta}) := p(\cdot; \boldsymbol{\theta}) / p(S; \boldsymbol{\theta})$, and then computing the average of the resulting gradients,

$$\nabla_{\boldsymbol{\theta}} \log p(S; \boldsymbol{\theta}) \approx \frac{1}{M} \sum_{i=1}^{M} \nabla_{\boldsymbol{\theta}} \log p(\sigma^{(i)}; \boldsymbol{\theta}). \tag{7}$$

Conceptually similar approaches have been used to approximate the gradient of the log-normalizer when maximizing the likelihood of energy-based probabilistic models (Song & Kingma, 2021).

It remains to show how to obtain samples from $p(\cdot \mid S; \boldsymbol{\theta})$, which we do by using a Markov chain Monte Carlo method on $\mathcal{S}_S$. More concretely, we employ a Metropolis-Hastings chain (Levin et al., 2009); at each time step, given the current permutation $\sigma$, the chain proposes a new permutation $\sigma_{\text{new}}$ according to proposal distribution $Q(\sigma_{\text{new}} \mid \sigma)$, and transitions to $\sigma_{\text{new}}$ with probability

$$p_{\text{accept}} = \min \left( 1, \frac{p(\sigma_{\text{new}} \mid S; \boldsymbol{\theta}) \, Q(\sigma \mid \sigma_{\text{new}}; \boldsymbol{\theta})}{p(\sigma \mid S; \boldsymbol{\theta}) \, Q(\sigma_{\text{new}} \mid \sigma; \boldsymbol{\theta})} \right).$$

A simple choice for $Q$ is the uniform distribution over all permutations regardless of the current state. We instead employ a more sophisticated proposal that is based on the form of the marginal probability (6), and leads to faster and more stable learning in practice. While a detailed discussion and theoretical analysis of mixing is beyond the scope of this paper, in Appendix E we present our proposal in detail, and compare it to the uniform.

## 5 Further related work

There has been a long line of research focused on understanding and reconstructing the evolutionary history of tumors (Nik-Zainal et al., 2012; Welch et al., 2012; Turajlic et al., 2015; Mitchell

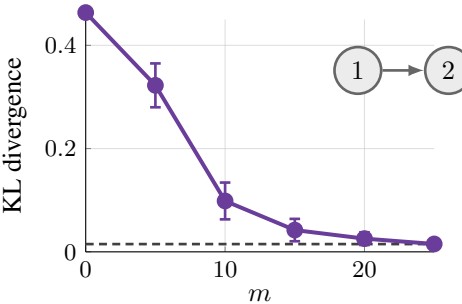
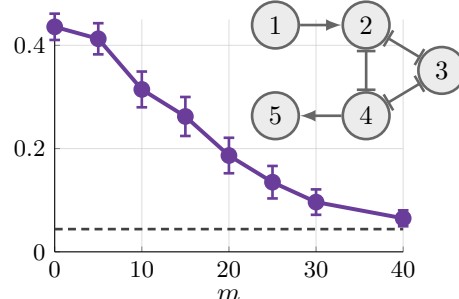

Figure 2: The KL divergence of recovered vs. true model as a function of the number $m$ of extra independent items, evaluated on two synthetic examples with known true parameters. The dashed lines represent the learned models given the true observations times. Error bars are over 96 repetitions.

et al., 2018; Jolly & Van Loo, 2018). However, these papers do not attempt to systematically model the timed interaction of genetic alterations across a data set. More recently, Gerstung et al. (2020) showed that modern data resulting from deep sequencing procedures may contain partial order information about specific genetic events. Incorporating this additional information into an analysis like ours, could potentially help with further reducing underspecification. Finally, there is evidence that specific classes of mutational processes generate mutations at a constant rate, and are largely unaffected by other factors (Alexandrov et al., 2015; Campbell et al., 2020; Alexandrov et al., 2020). The resulting so-called mutational signatures have been likened to a biological clock, and could be candidates for explicitly defining the set $V_+$ discussed earlier (cf. Section 3.2).

# 6 Experiments

The following experimental setup is common to all our experiments. To optimize the objective (3), we use a proximal AdaGrad method (Duchi et al., 2011). We fix the initial step size to $\eta = 1$, and the regularization weight to $\lambda = 0.01$. To initialize the parameter matrix $\boldsymbol{\Theta}$, we train a diagonal model for 50 epochs, and then draw each off-diagonal entry from $\mathrm{Unif}([-0.2, 0.2])$. For the gradient approximation (7), we use $M = 50$ samples in addition to 10 burn-in samples that are discarded. The code used to run our experiments can be found at `https://github.com/3lectrologos/time/tree/clean`.

## 6.1 Synthetic data

We start by evaluating our approach on two synthetic data sets drawn from known true models. The first data set is identical to the one discussed in Figure 1, and contains observations of two items that tend to follow a specific time order (item 1 comes before item 2). The second data set contains observations of five items following the graphical structure shown on the upper right of Figure 2. Edges with arrows $\leftarrow$ represent positive interactions ($\theta_{ij} > 0$), while edges with vertical lines $\vdash$ represent negative interactions ($\theta_{ij} < 0$). This type of structure, containing groups of items with bidirectional negative interactions, is common in genomics data; in fact, this exact structure can be found in the model learned from real cancer data in our following experiment (see Appendix F).

Our goal is to evaluate how well we can recover the true model by optimizing the regularized marginal likelihood; in particular, we want to examine the effect of adding a number $m$ of independent items to the data. To quantify the quality of recovery, we compare the true distribution $p_{\text{true}}$ of marginal sequences, computed using eq. (6), to the distribution of marginal sequences $p^*$ of the learned model, approximated by drawing $10^6$ marginal sequences from that model. In Figure 2 we depict the KL divergence $d_{\text{KL}}(p^* \,\|\, p_{\text{true}})$ for increasing numbers $m$ of extra items. To compare against the ideal case, we also train a model by maximizing the regularized marginal likelihood given the true observation times (cf. Section 4.2). The KL divergence between this model and the true one is shown as a dashed line in the figure. We repeated each experiment 96 times; sources of randomness include the choice of independent items, the parameter initialization, and the gradient approximation. We see that the recovery quality improves for both data sets with increasing $m$, and rapidly approaches the ideal. This shows that observing a large enough number of additional items is practically equivalent to knowing the true observation times, and thus validates Theorem 1.

Table 1: Learning run times (Intel Core i9 CPU)

| Method | $n = 10$ | $n = 15$ | $n = 20$ | $n = 50$ | $n = 100$ |
|---|---|---|---|---|---|
| Schill et al. (2019) | 2 s | 43 s | 121 m | – | – |
| Ours | 3 s | 4 s | 8 s | 1 m 5 s | 33 m 43 s |

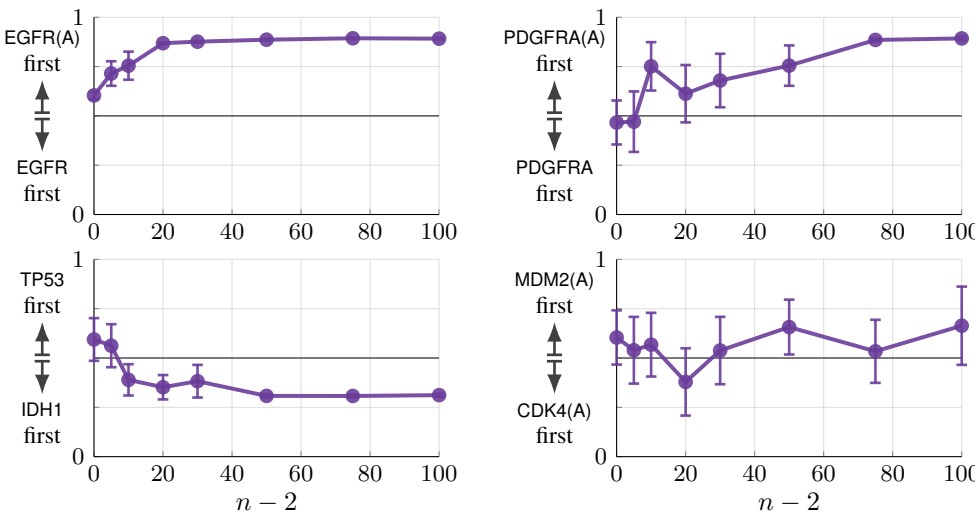

Figure 3: Results on the TCGA glioblastoma data set indicating the tendencies in the inferred time order of four selected pairs of genetic alterations. Error bars are over 24 repetitions.

## 6.2 Real cancer data

We next evaluate our approach on a data set that contains $N = 378$ tumor samples of glioblastoma multiforme, an aggressive type of brain cancer. The data is part of the TCGA PanCancer Atlas project (`https://www.cancer.gov/tcga`), and we obtained a preprocessed version via cBioPortal (Cerami et al., 2012). We followed some further filtering procedures from previous work (Leiserson et al., 2013) and ended up with a ground set of $n = 410$ items, each of which represents a point mutation, amplification, or deletion of a specific gene. Analyzing the interaction structure between such genetic alterations is of fundamental importance to cancer research, as it can help illuminate the processes that are involved in cancer initiation and progression.

For the remainder of this section, we assume that the items in $V$ are ordered by decreasing frequency, and when selecting a subset we keep the most frequent items. In Table 1, we show that our learning approach runs about 1000 times faster on a ground set of size $n = 20$ compared to the exponentially scaling exact approach proposed by Schill et al. (2019).

In the absence of ground truth, we use the following procedure to evaluate how some of the learned parameters behave as we increase the size of the ground set. Given a pair of items $a, b \in V$, and $n-2$ other (most frequent) items, we first learn a CTMC model on the $n$ total items using our approach. Next we approximate the sequence distribution of just $a$ and $b$ (marginally over time and over all other items) by sampling $10^6$ marginal sequences from the learned model. Finally, we compute the proportion of sequences in which $a$ occurred before $b$, given that both where observed; when this proportion is greater than $0.5$, we infer that $a$ is more likely to occur before $b$, and vice versa. In Figure 3 we present these computed proportions for four chosen pairs of genetic alterations that are interesting both from a biological standpoint, as well as in terms of learning behavior.

First, we discover two significant ordered interactions, shown in the top row of the figure, which were not reported in previous work analyzing glioblastoma data (Raphael & Vandin, 2015; Cristea et al., 2017; Schill et al., 2019). Our results suggest that amplification events for genes EGFR and PDGFRA tend to occur before their respective point mutation events. Interestingly, the co-occurrence of amplifications and mutations for EGFR has been observed before (Leiserson et al.,

2013; Sanchez-Vega et al., 2018), but there is little known about their time order. Note that the time order of the PDGFRA alterations can be robustly inferred only after including more than 70 additional items in the analysis.

Second, we confirm some previously reported interactions, for example, the tendency of IDH1 mutations to occur before TP53 ones, as shown in the bottom left of the figure. This was observed before by Schill et al. (2019), and is also supported by some biological evidence (Watanabe et al., 2009). Again, we can see that our result becomes robust only after $n = 50$. This indicates that the previous observation of this time order may have been in part due to a fortuitous choice of the optimization setup, e.g., fixed initialization (see Appendix G). The final result in the bottom right of the figure seems to support this claim: the amplifications of MDM2 and CDK4 do not seem to have a clear time order, even though previous work has inferred both that CDK4(A) occurs before MDM2(A) (Cristea et al., 2017), as well as the opposite order (Schill et al., 2019).

Finally, the parameter matrix $\Theta$ learned from this data set (see Appendix G) has indeed an approximately block-diagonal structure, which confirms the practical relevance of Theorem 1.

## 7   Conclusion

Our experiments demonstrated that, in the presence of underspecification, special care is required when interpreting results that may be significantly affected by particular choices in the optimization setup. In this paper, we proposed an effective way to mitigate such effects, namely scaling up the problem and taking into account additional items that are commonly discarded as insignificant. We hope that our approach will inspire further work into combating underspecification and learning robust time evolution models.

## Acknowledgments and Disclosure of Funding

This work was partially supported by a Swiss National Science Foundation Early Postdoc.Mobility fellowship, by NSF CAREER award 1553284, NSF BIGDATA IIS-1741341, and NSF award 1717610, as well as a grant from the US National Cancer Institute (1R35CA220523).

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
