# A  Proof of Proposition 1

**Proposition 1.** *There is a one-dimensional family of parameters $\boldsymbol{\theta} = \boldsymbol{\theta}(s)$, and $s_1, s_2 \in \mathbb{R}$, such that $p(\cdot\,;\boldsymbol{\theta}(s)) = p(\cdot\,;\boldsymbol{\theta}^*)$, for all $s \in (s_1, s_2)$.*

*Proof.* We use (6) to write down the following probabilities,

$$p_0(\boldsymbol{\theta}) := p(\emptyset; \boldsymbol{\theta}) = p((); \boldsymbol{\theta}) = \frac{1}{1 + w_{11} + w_{22}}$$

$$p_1(\boldsymbol{\theta}) := p(\{1\}; \boldsymbol{\theta}) = p((1); \boldsymbol{\theta}) = \frac{w_{11}}{1 + w_{11} + w_{22}} \frac{1}{1 + w_{22}w_{12}} = \frac{p_0 w_{11}}{1 + w_{22}w_{12}}$$

$$p_2(\boldsymbol{\theta}) := p(\{2\}; \boldsymbol{\theta}) = p((2); \boldsymbol{\theta}) = \frac{w_{22}}{1 + w_{11} + w_{22}} \frac{1}{1 + w_{11}w_{21}} = \frac{p_0 w_{22}}{1 + w_{11}w_{21}}.$$

We define the probabilities of the true model $p_i^* := p_i(\boldsymbol{\theta}^*)$, for $i = 0, 1, 2$, and solve the system

$$p_i(\boldsymbol{\theta}) = p_i^*, \quad i \in \{0, 1, 2\}$$
$$w_{ij} > 0, \quad i, j \in \{1, 2\}.$$

The solution is the following parametric family,

$$w_{11} = s$$
$$w_{22} = 1/p_0^* - 1 - s$$
$$w_{21} = \frac{1 - p_0^* - p_2^* - p_0^* s}{p_2^* s}$$
$$w_{12} = \frac{p_0^* s - p_1^*}{p_1^*(1/p_0^* - 1 - s)},$$

for $s \in (s_1, s_2)$, where $s_1 = p_1^*/p_0^*$ and $s_2 = (1 - p_0^* - p_2^*)/p_0^*$. Then, the parameter family $\boldsymbol{\theta} = \boldsymbol{\theta}(s)$ can be computed as $\theta_{ij}(s) = \log w_{ij}(s)$, for $i, j \in \{1, 2\}$.  $\square$

**Illustration of the parameter family**

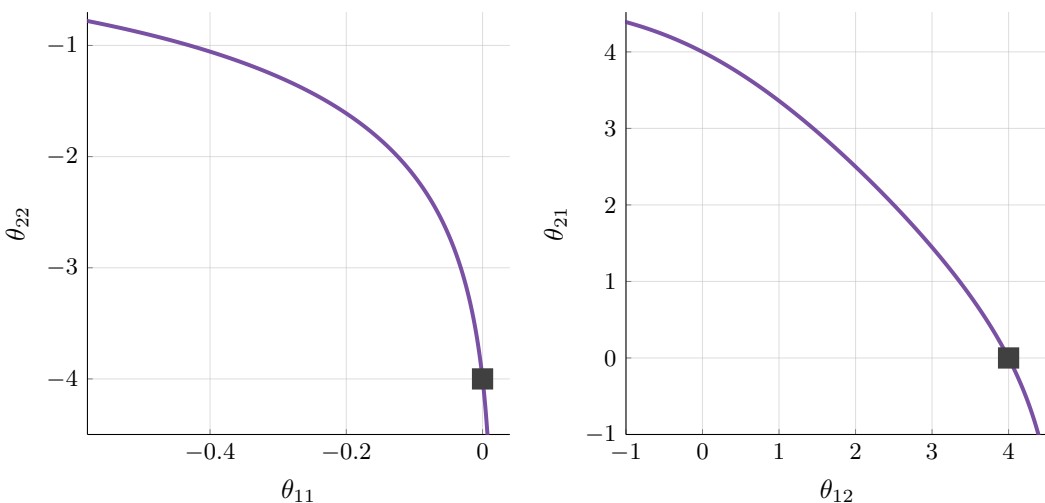

Figure 4: The family of parameters that result in the same marginal distribution of sets $p(S; \boldsymbol{\theta}^*)$ for the two-item example discussed in Section 3 and for $\alpha = 4$. The square marks highlight the true model parameters $\theta_{11}^* = 0, \theta_{22}^* = -\alpha, \theta_{12}^* = \alpha, \theta_{21}^* = 0$.

# B    Proof of Theorem 1

We consider a CTMC with parameter matrix $\mathbf{\Theta}_+ = \theta_+ \mathbf{I}_m$. As a reminder, for simplicity of notation we define $w_+ = e^{\theta_+}$. The following lemma derives the mean and variance of the posterior time distribution, for any fixed observation $S_+$.

**Lemma 1.** *For any $m > 0$, and any observation $S_+ \subseteq V$ with $|S_+| = k$, if we define $r = k/m$, then the posterior time distribution $p(t \mid S_+; \boldsymbol{\theta}_+)$ has mean and variance given by*

$$M_{post} = \frac{1}{w_+}(\psi(\alpha + \beta) - \psi(\alpha))$$

$$V_{post} = \frac{1}{w_+^2}(\psi_1(\alpha) - \psi_1(\alpha + \beta)),$$

*where $\alpha = 1/w_+ + (1-r)m + 1$, $\beta = rm + 1$, and $\psi$, $\psi_1$ are the digamma and trigamma functions respectively.*

*Proof.* At any time $t \geq 0$, and for any $x \in V$, if we denote by $X$ the randomly observed set, then the indicator random variable $[\![x \in X]\!]$ is Bernoulli with success probability $1 - e^{-tw_+}$. Therefore, the variable $|X|$ follows a binomial distribution with parameters $m$ and $1 - e^{-tw_+}$, and the posterior time density can be written as

$$\begin{aligned}
p(t \mid S_+; \boldsymbol{\theta}_+) &= p(t \mid |X| = k; \boldsymbol{\theta}_+) \\
&\propto p(|X| = k \mid t; \boldsymbol{\theta}_+)p(t) \\
&= \left(1 - e^{-tw_+}\right)^k \left(e^{-tw_+}\right)^{m-k} e^{-t} \\
&= \left(1 - e^{-tw_+}\right)^{rm} e^{-t(1 + (1-r)w_+ m)}.
\end{aligned}$$

Consider now the variable transformation $y = e^{-tw_+}$ with derivative $\mathrm{d}y/\mathrm{d}t = -w_+ e^{-tw_+}$. If $T$ is a random variable with density $p(\cdot \mid S_+; \boldsymbol{\theta}_+)$, then the random variable $Y = e^{-Tw_+}$ has density given by

$$p(y \mid S_+; \boldsymbol{\theta}) \propto y^{1/w_+ + (1-r)m}(1-y)^{rm},$$

that is, $Y \sim \mathrm{Beta}(\alpha, \beta)$, where $\alpha = 1/w_+ + (1-r)m + 1$, and $\beta = rm + 1$. We can then write the posterior time mean and variance as

$$M_{\mathrm{post}} = \mathbb{E}[T] = -\frac{1}{w_+}\mathbb{E}[\log Y]$$

$$V_{\mathrm{post}} = \mathrm{Var}[T] = \frac{1}{w_+^2}\mathrm{Var}[\log Y].$$

The mean and variance of $\log Y$ can be computed with the help of the digamma function $\psi$, and trigamma function $\psi_1$, which are defined as

$$\psi(z) = \frac{\mathrm{d}}{\mathrm{d}z}\log\Gamma(z)$$

$$\psi_1(z) = \frac{\mathrm{d}^2}{\mathrm{d}^2 z}\log\Gamma(z),$$

where $\Gamma$ denotes the gamma function. We then have

$$M_{\mathrm{post}} = -\frac{1}{w_+}\mathbb{E}[\log Y] = \frac{1}{w_+}(\psi(\alpha + \beta) - \psi(\alpha))$$

$$V_{\mathrm{post}} = \frac{1}{w_+^2}\mathrm{Var}[\log Y] = \frac{1}{w_+^2}(\psi_1(\alpha) - \psi_1(\alpha + \beta)).$$

$\square$

Since the size $k$ of the observed set is randomly distributed given the observation time, the next lemma shows how this size concentrates around the mean.

**Lemma 2.** *Let $S_+ \subseteq V$ be randomly drawn according to the CTMC with parameter matrix $\Theta_+$, let $t^*$ be the true observation time, and define random variable $r = |S_+|/m$. Then, for any $\delta \in (0,1)$, and any $m \geq \sqrt{2/\delta}$, the following holds with probability at least $1 - \delta$,*

$$\left| r - (1 - e^{-w_+ t^*}) \right| \leq \sqrt{\frac{\log m}{m}}.$$

*Proof.* In the preceding lemma we saw that $|S_+|$ follows a binomial distribution with parameters $m$ and $1 - e^{-t^* w_+}$. Applying Hoeffdings's inequality gives us

$$\mathbb{P}\left[ \left| |S_+| - m(1 - e^{-w_+ t^*}) \right| \leq \sqrt{m \log m} \right] \geq 1 - \frac{2}{m^2}$$

$$\Rightarrow \mathbb{P}\left[ \left| r - (1 - e^{-w_+ t^*}) \right| \leq \sqrt{\frac{\log m}{m}} \right] \geq 1 - \frac{2}{m^2}.$$

$\square$

**Theorem 1.** *Let $S_+ \subseteq V$ be randomly drawn according to the CTMC with parameter matrix $\Theta_+$, and let $t^*$ be the true observation time of $S_+$. Then, for any $\delta \in (0,1)$, there exists a $m_0$, such that for all $m \geq m_0$, the mean and variance of the posterior time distribution $p(t \,|\, S_+; \boldsymbol{\theta})$ can be bounded as follows with probability at least $1 - \delta$,*

$$|M_{post} - t^*| \leq C_1(\theta_+, t^*)\sqrt{\frac{\log m}{m}} + \mathcal{O}\left(\frac{\log m}{m}\right)$$

$$V_{post} \leq C_2(\theta_+, t^*)\frac{1}{m} + \mathcal{O}\left(\frac{1}{m^2}\right).$$

*Proof.* We start with the posterior mean. By Lemma 1 we have

$$M_{\text{post}} = \frac{1}{w_+}(\psi(\alpha + \beta) - \psi(\alpha))$$

$$= \frac{1}{w_+}\left( \log(\alpha + \beta) - \frac{1}{2(\alpha + \beta)} - \log(\alpha) + \frac{1}{2\alpha} + \mathcal{O}\left(\frac{1}{(\alpha + \beta)^2}\right) + \mathcal{O}\left(\frac{1}{\alpha^2}\right) \right)$$

(by $\psi$ series expansion)

$$= \frac{1}{w_+}\left( \log\left(\frac{1}{w_+ m} + \frac{2}{m} + 1\right) - \log\left(\frac{1}{w_+ m} + (1 - r) + \frac{1}{m}\right) + \mathcal{O}\left(\frac{1}{m}\right) \right)$$

(by Lemma 2)

$$= -\frac{1}{w_+}\log(1 - r) + \mathcal{O}\left(\frac{1}{m}\right), \qquad \text{(by Taylor expansion of log)}$$

which holds with probability at least $1 - \delta$, for all $m \geq \max(\sqrt{2/\delta}, m_1)$, where $m_1$ is the smallest positive integer that satisfies $\left| \frac{1 + 2w_+}{w_+ m_1} \right| \leq 1$, and $\left| \frac{1 + w_+}{w_+ m_1} \right| \leq 1 - r$. These conditions are necessary for the Taylor expansions of the logarithms to be converging. Now we can bound the distance of the posterior mean from the true observation time as follows,

$$|M_{\text{post}} - t^*| = \left| -\frac{1}{w_+}\log(1 - r) + \frac{1}{w_+}\log\left(e^{-w_+ t^*}\right) + \mathcal{O}\left(\frac{1}{m}\right) \right|$$

$$= \frac{1}{w_+}\left| \log\left((1 - r)e^{-w_+ t^*}\right) \right| + \mathcal{O}\left(\frac{1}{m}\right)$$

$$= \frac{1}{w_+}\left| \log\left(1 + (1 - r - e^{-w_+ t^*})e^{w_+ t^*}\right) \right| + \mathcal{O}\left(\frac{1}{m}\right)$$

$$= \frac{1}{w_+}\left| (1 - r - e^{-w_+ t^*})e^{w_+ t^*} \right| + \mathcal{O}\left(\frac{1}{m}\right) + \mathcal{O}\left(\left(1 - r - e^{-w_+ t^*}\right)^2\right)$$

(by Taylor expansion of $\log(1 + \cdot)$)

$$\leq \frac{e^{w_+ t^*}}{w_+}\sqrt{\frac{\log m}{m}} + \mathcal{O}\left(\frac{\log m}{m}\right), \qquad \text{(by Lemma 2)}$$

which holds with probability at least $1 - \delta$, for all $m \geq \max(\sqrt{2/\delta}, m_1, m_2)$, where $m_2$ is the smallest positive integer that satisfies $\left|(1 - r - e^{-w_+ t^*})e^{w_+ t^*}\right| \leq 1$. Again, this condition is necessary for the Taylor expansion of the logarithm to be converging. The bound of the theorem follows by defining $C_1(w_+, t^*) := \dfrac{e^{w_+ t^*}}{w_+}$.

The argument for the posterior variance follows a similar structure. By Lemma 1 we have

$$
\begin{aligned}
V_{\text{post}} &= \frac{1}{w_+^2}(\psi_1(\alpha) - \psi_1(\alpha + \beta)) \\
&= \frac{1}{w_+^2}\left(\frac{1}{\alpha} - \frac{1}{\alpha + \beta} + \mathcal{O}\left(\frac{1}{(\alpha + \beta)^2}\right) + \mathcal{O}\left(\frac{1}{\alpha^2}\right)\right) \quad \text{(by $\psi_1$ series expansion)} \\
&= \frac{r}{w_+^2(1 - r)}\frac{1}{m} + \mathcal{O}\left(\frac{1}{m^2}\right).
\end{aligned}
$$

Using Lemma 2, we can show that $r/(1 - r)$ is upper bounded by a constant in $m$ with probability at least $1 - \delta$. In particular, for all $m \geq \max(\sqrt{2/\delta}), m_3$, where $m_3 := 1/(1 - e^{-2w_+ t^*})$, we get $1 - r \geq e^{-w_+ t^*}/2 > 0$. It follows that

$$
\frac{r}{1 - r} \leq \frac{2 - e^{-w_+ t^*}}{e^{-w_+ t^*}}.
$$

The posterior variance bound follows by defining $C_2(w_+, t^*) := \dfrac{2 - e^{-w_+ t^*}}{w_+^2 e^{-w_+ t^*}}$.

To conclude the formulation of the theorem, we define $m_0 := \max(\sqrt{2/\delta}, m_1, m_2, m_3)$. $\qquad \square$

**Illustration of $C_1$ and $C_2$**

The following figure shows the constants $C_1(w_+, t^*)$ and $C_2(w_+, t^*)$ for different values of $w_+$ and $t^*$. Note that as we keep increasing the true observation time $t^*$, the values of $w_+$ that minimize $C_1$ and $C_2$ keep decreasing. This verifies the intuition provided in the main text that larger rates for the independent elements in $V_+$, that is, larger values of $w_+$ will be more suitable (i.e., will have faster convergence in $m$) for determining smaller observation times $t^*$ and vice versa. We also see that larger times are in general harder to estimate as evidenced by the increase of the minimum values of $C_1(\cdot, t^*)$ and $C_2(\cdot, t^*)$ with increasing $t^*$.

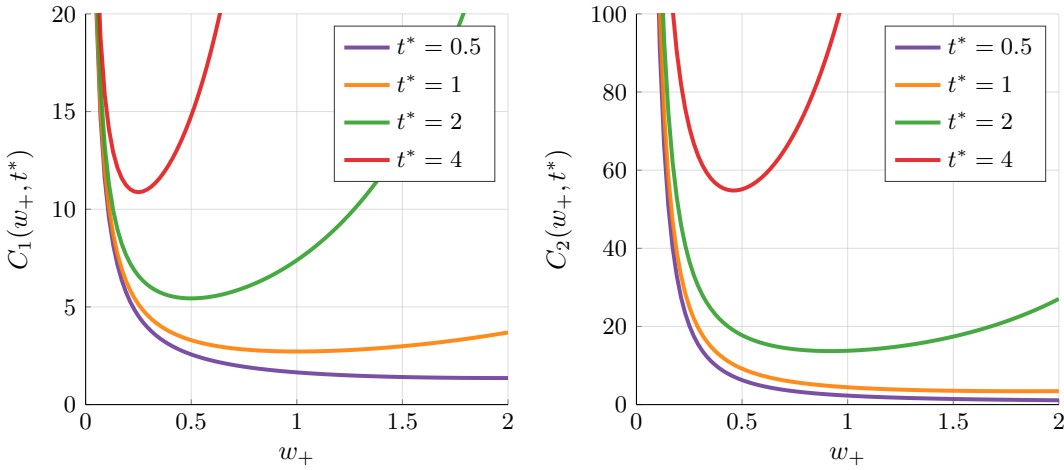

Figure 5: The terms $C_1$ and $C_2$ that appear in the bounds of Theorem 1, plotted for different values of $w_+$ and $t^*$.

## C  Violating the assumptions of Theorem 1

We show that the time posterior can behave in a similar way to what is described in Theorem 1 even when the assumption about $\boldsymbol{\Theta}_+$ having the form $\theta_+\boldsymbol{I}_m$ is relaxed.

In a similar setup to Figure 1, given a matrix $\boldsymbol{\Theta}_+$, we draw $N = 1000$ samples and compute the variance of the posterior time distribution for each sample. We then plot the average variance over the samples for each choice of $\boldsymbol{\Theta}_+$.

Three of the lines in Figure 6 satisfy the assumptions of Theorem 1 each with a different choice of $\theta_+$. The fourth line corresponds to independently drawing each diagonal entry of $\boldsymbol{\Theta}_+$ uniformly from $[-3, -1]$. Both axes are plotted in logarithmic scale, and the error bars denote two standard errors of the mean. The dashed line denotes the function $c/m$ for some constant $c$, and is provided for reference. Note that the setting where the diagonal parameters are drawn uniformly exhibits similar behavior to the constant diagonal setting. In particular, we can see that in all cases the variances asymptotically decrease with rate $1/m$, as shown in our theorem.

Next, we assume that $\boldsymbol{\Theta}_+$ is block-diagonal with blocks of size 2, and introduce positive off-diagonal parameters $\gamma$ within each block. In other words, we violate the independence assumption of Theorem 1, and introduce increasingly attractive behavior between pairs of elements. To make sure that we are able to compare the different choices of $\gamma$ on equal grounds, we adjust the diagonal parameters of $\boldsymbol{\Theta}_+$, so that the marginal frequency of each item in $S_+$ stays constant as we vary $\gamma$. In Figure 7, we see that the posterior variance increases up to some point with increasing interaction strength. Intuitively, we expect that for high enough $\gamma$, the two elements of each block are practically co-occurring (i.e., either neither or both are observed), and therefore the effect of each block on the time posterior is reduced to the effect of a single element. Figure 7 confirms this intuition; for example, the average variance for $m = 100$ and $\gamma = 12$ can be seen to be roughly the same as the average variance for $m = 50$ and $\gamma = 0$ (independent case).

In Figure 8, we show analogous results for $\gamma < 0$, that is, repulsive interaction between pairs of items in $S_+$. In this case, the effect on the posterior time estimates seems to be less noticeable compared to the attractive case.

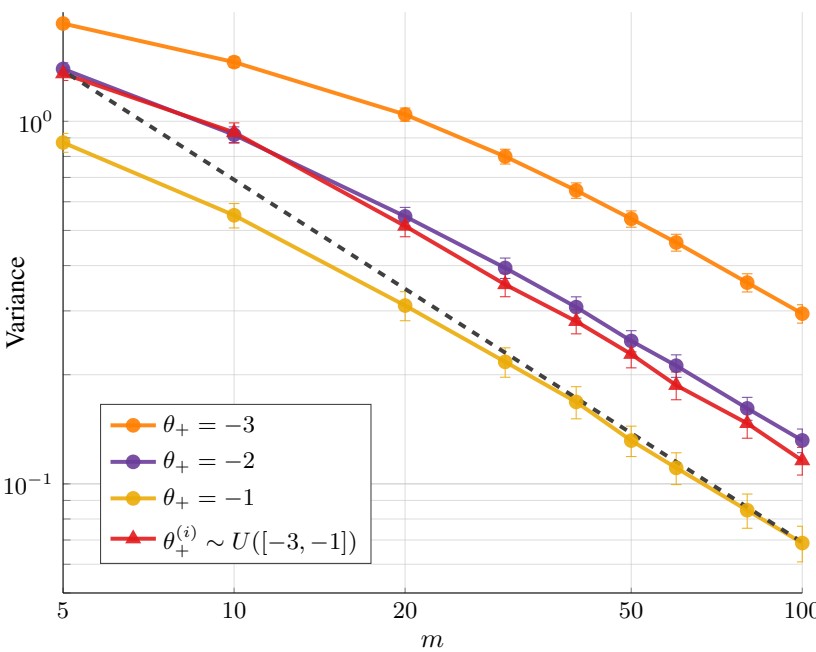

Figure 6: The average variance of the time posteriors for different choices of diagonal $\boldsymbol{\Theta}_+$.

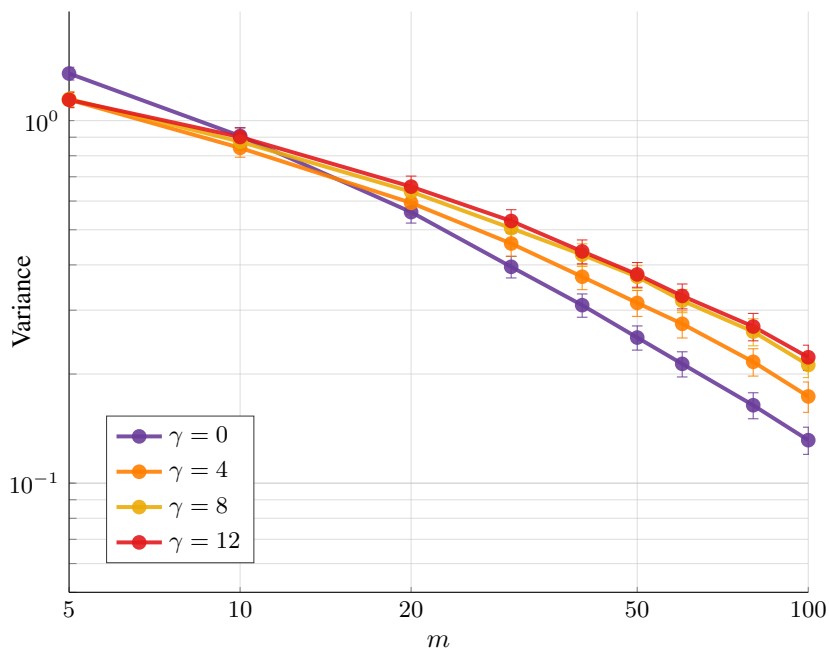

Figure 7: The average variance of the time posteriors for different strengths of attractive interaction. The independent case $\gamma = 0$ is equivalent to the $\theta_+ = -2$ case in Figure 6.

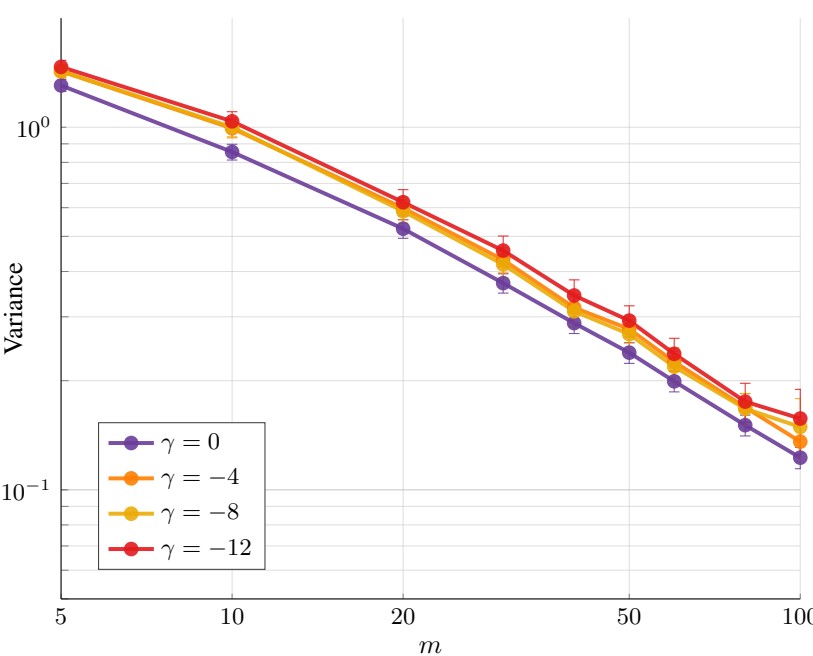

Figure 8: The average variance of the time posteriors for different strengths of repulsive interaction. The independent case $\gamma = 0$ is equivalent to the $\theta_+ = -2$ case in Figure 6.

# D    Proof of Proposition 2

**Proposition 2.** *The marginal probability of a partial sequence* $\sigma = (\sigma_1, \ldots, \sigma_k)$ *can be written as*

$$p(\sigma; \boldsymbol{\theta}) = \left( \prod_{i=1}^{k} \frac{q_{\sigma_{[i-1]} \to \sigma_{[i]}}(\boldsymbol{\theta})}{1 + \tilde{q}_{\sigma_{[i-1]}}(\boldsymbol{\theta})} \right) \frac{1}{1 + \tilde{q}_{\sigma_{[k]}}(\boldsymbol{\theta})}.$$

*Proof.* Event $\mathcal{B}$ is defined by $T_k < T_{\text{obs}} < T_{k+1}$ To compute $\mathbb{P}(\mathcal{B} \mid \mathcal{A})$ we focus first on the event defined by one of the above inequalities, and rewrite its probability using the definition of jump times and the chain rule as follows,

$$\mathbb{P}(T_{\text{obs}} > T_k \mid \mathcal{A}) = \mathbb{P}\left( T_{\text{obs}} > \sum_{i=1}^{k} H_i \,\middle|\, \mathcal{A} \right)$$

$$= \prod_{j=1}^{k} \mathbb{P}\left( T_{\text{obs}} > \sum_{i=1}^{j} H_i \,\middle|\, T_{\text{obs}} > \sum_{i=1}^{j-1} H_i, \mathcal{A} \right).$$

The crucial observation here is that we can greatly simplify the above expression by making use of the memoryless property of exponential random variables (Bertsekas & Tsitsiklis, 2008). According to this property, if $Z$ is an exponential random variable, then $\mathbb{P}(Z > x + y \mid Z > x) = P(Z > y)$, for all $x, y \in \mathbb{R}$. The above expression then becomes

$$\mathbb{P}(T_{\text{obs}} > T_k \mid \mathcal{A}) = \prod_{j=1}^{k} \mathbb{P}(T_{\text{obs}} > H_j \mid \mathcal{A})$$

$$= \prod_{j=1}^{k} \mathbb{P}(H_j - T_{\text{obs}} < 0 \mid \mathcal{A})$$

$$= \prod_{j=1}^{k} \frac{\tilde{q}_{\sigma_{[j-1]}}(\boldsymbol{\theta})}{1 + \tilde{q}_{\sigma_{[j-1]}}(\boldsymbol{\theta})}. \tag{8}$$

The last equality comes from the fact that the distribution of the difference of two independent exponential random variables with rates $\lambda_1, \lambda_2$ is an asymmetric Laplace distribution with parameters $\lambda_2$ and $\lambda_1$ on the negative and positive half-lines respectively.

An analogous derivation gives us

$$\mathbb{P}(T_{\text{obs}} < T_{k+1} \mid T_{\text{obs}} > T_k, \mathcal{A}) = 1 - \mathbb{P}(T_{\text{obs}} > T_{k+1} \mid T_{\text{obs}} > T_k, \mathcal{A})$$

$$= 1 - \mathbb{P}(T_{\text{obs}} > H_{k+1} \mid \mathcal{A}) \qquad \text{(by memorylessness)}$$

$$= 1 - \frac{\tilde{q}_{\sigma_{[k]}}(\boldsymbol{\theta})}{1 + \tilde{q}_{\sigma_{[k]}}(\boldsymbol{\theta})} \qquad \text{(by difference of exp.)}$$

$$= \frac{1}{1 + \tilde{q}_{\sigma_{[k]}}(\boldsymbol{\theta})}. \tag{9}$$

Finally, combining the results of (8) and (9), we get the probability of a partial sequence,

$$p(\sigma; \boldsymbol{\theta}) = \mathbb{P}(\mathcal{A}) \mathbb{P}(\mathcal{B} \mid \mathcal{A})$$

$$= \mathbb{P}(\mathcal{A}) \mathbb{P}(T_{\text{obs}} > T_k \mid \mathcal{A}) \mathbb{P}(T_{\text{obs}} < T_{k+1} \mid T_{\text{obs}} > T_k, \mathcal{A})$$

$$= \left( \prod_{i=1}^{k} \frac{q_{\sigma_{[i-1]} \to \sigma_{[i]}}(\boldsymbol{\theta})}{1 + \tilde{q}_{\sigma_{[i-1]}}(\boldsymbol{\theta})} \right) \frac{1}{1 + \tilde{q}_{\sigma_{[k]}}(\boldsymbol{\theta})}.$$

$\square$

## E MCMC proposal

As a reminder, we would like to draw samples from $p(\cdot \mid S; \boldsymbol{\theta})$, which we do by using a Metropolis-Hastings chain over state space $\mathcal{S}_S$. At each time step, given the current permutation $\sigma$, the chain proposes a new permutation $\sigma_{\text{new}}$ according to proposal distribution $Q(\sigma_{\text{new}} \mid \sigma)$, and transitions to $\sigma_{\text{new}}$ with probability

$$p_{\text{accept}} = \min\left(1, \frac{p(\sigma_{\text{new}} \mid S; \boldsymbol{\theta})\, Q(\sigma \mid \sigma_{\text{new}}; \boldsymbol{\theta})}{p(\sigma \mid S; \boldsymbol{\theta})\, Q(\sigma_{\text{new}} \mid \sigma; \boldsymbol{\theta})}\right). \tag{10}$$

We focus on proposal distributions $Q(\sigma_{\text{new}} \mid \sigma; \boldsymbol{\theta}) = Q(\sigma_{\text{new}}; \boldsymbol{\theta})$ that do not depend on the current state $\sigma$. A simple such choice is the uniform proposal $Q(\sigma_{\text{new}}) = 1/|S|!$. In general, the convergence rate of such proposals depends crucially on the minimum ratio $Q(\cdot)/p(\cdot \mid S; \boldsymbol{\theta})$ over all states (Mengersen & Tweedie, 1996); intuitively, we want the proposal to match as well as possible the true distribution, and, in particular, to have a high chance to propose states that have high probability according to $p(\cdot \mid S; \boldsymbol{\theta})$.

Algorithm 2 shows how to draw a permutation $\sigma$ from our proposal $Q$, and at the same time compute its unnormalized probability $Q_{\text{val}} \propto Q(\sigma; \boldsymbol{\theta})$. This is enough to use our proposal in the Metropolis-Hastings chain, since the acceptance probability in (10) only requires computing ratios $Q(\sigma; \boldsymbol{\theta})/Q(\sigma_{\text{new}}; \boldsymbol{\theta})$. As shown in the algorithm, given a set $S$, we iteratively add items to $\sigma$ by considering at each step a weight $u_\nu$ for each candidate item $\nu \in S \setminus \sigma$. The form of $u_\nu$ bears a strong similarity to the form of the marginal sequence probability derived in (6). In fact, it is easy to show that for $|V| = 2$ this algorithm samples exactly from the correct distribution, that is, $Q(\cdot) = p(\cdot \mid S; \boldsymbol{\theta})$. More generally, the factor $c_\nu$ approximates the denominator of eq. (6), and intuitively takes into account the interactions between the elements that are already in $\sigma$ and all other elements in $V$. The factor $\prod w_{\nu j}$ computes the numerator of eq. (6), and intuitively takes into account the interactions that would occur between item $\nu$ (were we to add this item at this position) and all remaining items in $S$ that would be added subsequently.

To compare this proposal to the uniform one, we use the same synthetic data set as in Figure 1, but constrain the size of the ground set to be at most $n = 20$, so that we can compare to the exact gradients. We first run 100 gradient ascent epochs using the exact gradients, and then approximate the gradient of the marginal likelihood at that point using the Metropolis-Hastings sampler with the two different proposals. If we denote by $\boldsymbol{g}$ the true gradient, and by $\hat{\boldsymbol{g}}$ the gradient approximation, Figure 9 plots the error $\|\hat{\boldsymbol{g}} - \boldsymbol{g}\|_2$ as a function of the number of samples used. Note that for $n = 10$ the two proposals seem to perform similarly, but as $n$ gets larger the advantage of our proposal becomes increasingly more pronounced. In practice we have observed that for larger ground sets our proposal tends to provide considerable improvements to the convergence speed of the optimization algorithm, and also be much less sensitive to the choice of step size compared to the uniform proposal.

---

**Algorithm 2:** Drawing from proposal $Q$

**Input :** Parameters $\boldsymbol{\theta}$, set $S$

$Q_{\text{val}} \leftarrow 0$

$\sigma \leftarrow ()$

**for** $k = 1, \ldots, |S|$ **do**

    **for** $\nu \in S \setminus \sigma$ **do**

        $\sigma' \leftarrow \sigma \cup \{\nu\}$

        $d_\nu \leftarrow 1 + \sum_{j \in V \setminus \sigma'} w_{jj} \prod_{i \in \sigma'} w_{ij}$

        $u_\nu \leftarrow \dfrac{1}{d_\nu} \prod_{j \in S \setminus \sigma'} w_{\nu j}$

    Draw $x \sim \text{Cat}(S \setminus \sigma, (u_\nu / \sum_j u_j)_{\nu \in S \setminus \sigma})$

    $\sigma \leftarrow \sigma \,\|\, x$

    $Q_{\text{val}} \leftarrow Q_{\text{val}}\, u_x / \sum_j v_j$

**return** $\sigma, Q_{\text{val}}$

---

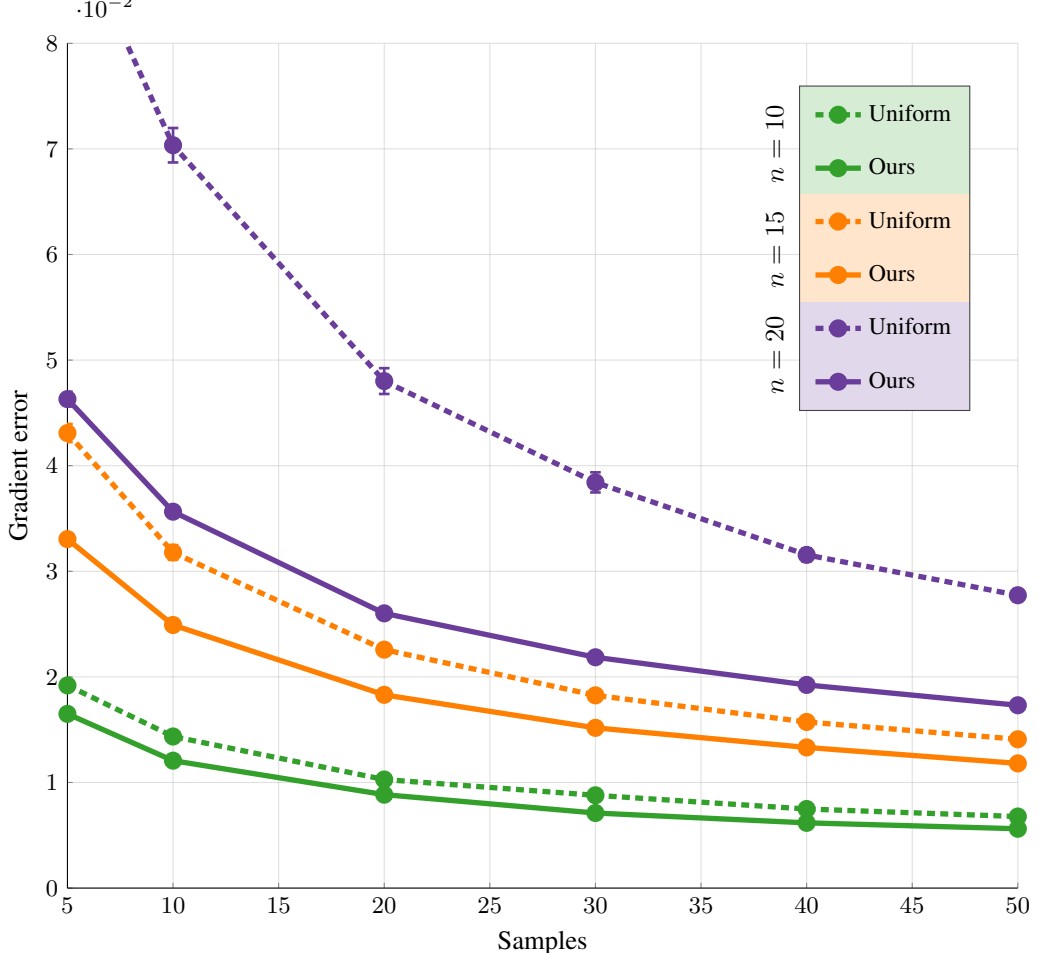

Figure 9: The norm of the difference between the true gradient and the gradient approximation using sampling. Dashed lines correspond to the uniform proposal, while solid lines correspond to our proposal described in Algorithm 2. Each color denotes a different ground set size.

## F    Synthetic data sets

**Two-item data set**

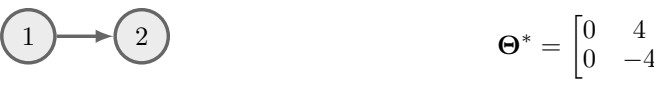

$$\mathbf{\Theta}^* = \begin{bmatrix} 0 & 4 \\ 0 & -4 \end{bmatrix}$$

**Five-item data set**

$$\mathbf{\Theta}^* = \begin{bmatrix} -1 & 4 & 0 & 0 & 0 \\ 0 & -1 & -2 & -2 & 0 \\ 0 & -2 & -1 & -2 & 0 \\ 0 & -2 & -2 & -0.5 & 4 \\ 0 & 0 & 0 & 0 & -4 \end{bmatrix}$$

# G   Further experimental results

Figure 10 shows results on the TCGA glioblastoma data set used by Schill et al. (2019), and originally preprocessed by Leiserson et al. (2013). (This is an older version of the data used in our experiments.) We run our method on the same subset of $n = 20$ genetic alterations chosen by Schill et al. (2019). The two plots in the top row show the learned parameter matrices $\Theta$ for two different random initializations. The left matrix is practically identical to the result reported by Schill et al. (2019), while the right matrix has some notable differences. To further highlight these differences, the bottom plot shows the range (max - min value) of each learned parameter across 20 random initializations.

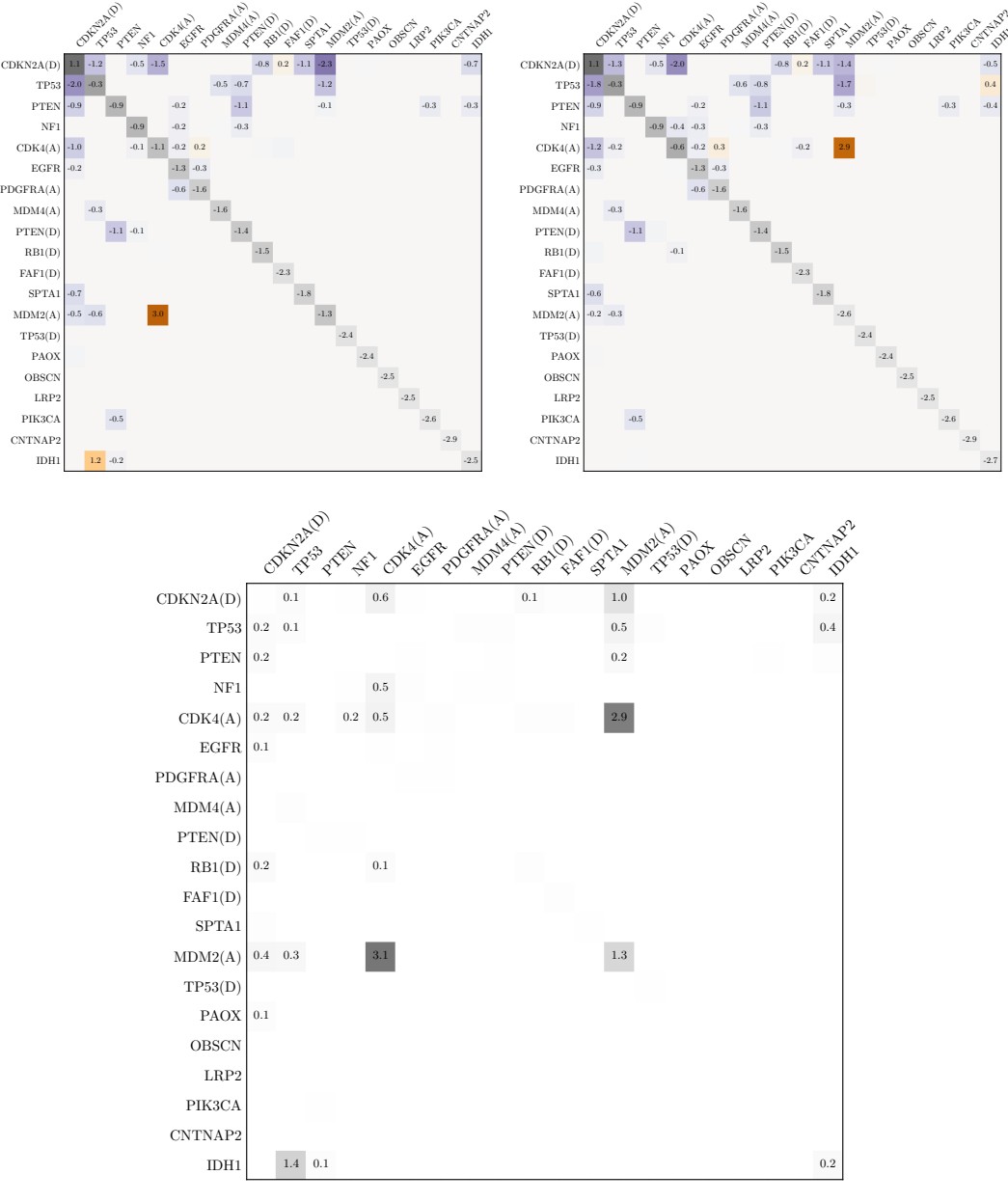

Figure 10: Top: Learned parameter matrices $\Theta$ on the reduced data set ($n = 20$) used by Schill et al. (2019) for two different random initializations. Orange shades denote $\theta_{ij} > 0$, while purple shades denote $\theta_{ij} < 0$. The zero entries are left blank. Bottom: Learned parameter ranges across 20 random initializations.

**Approximately block-diagonal interaction structure in real data**

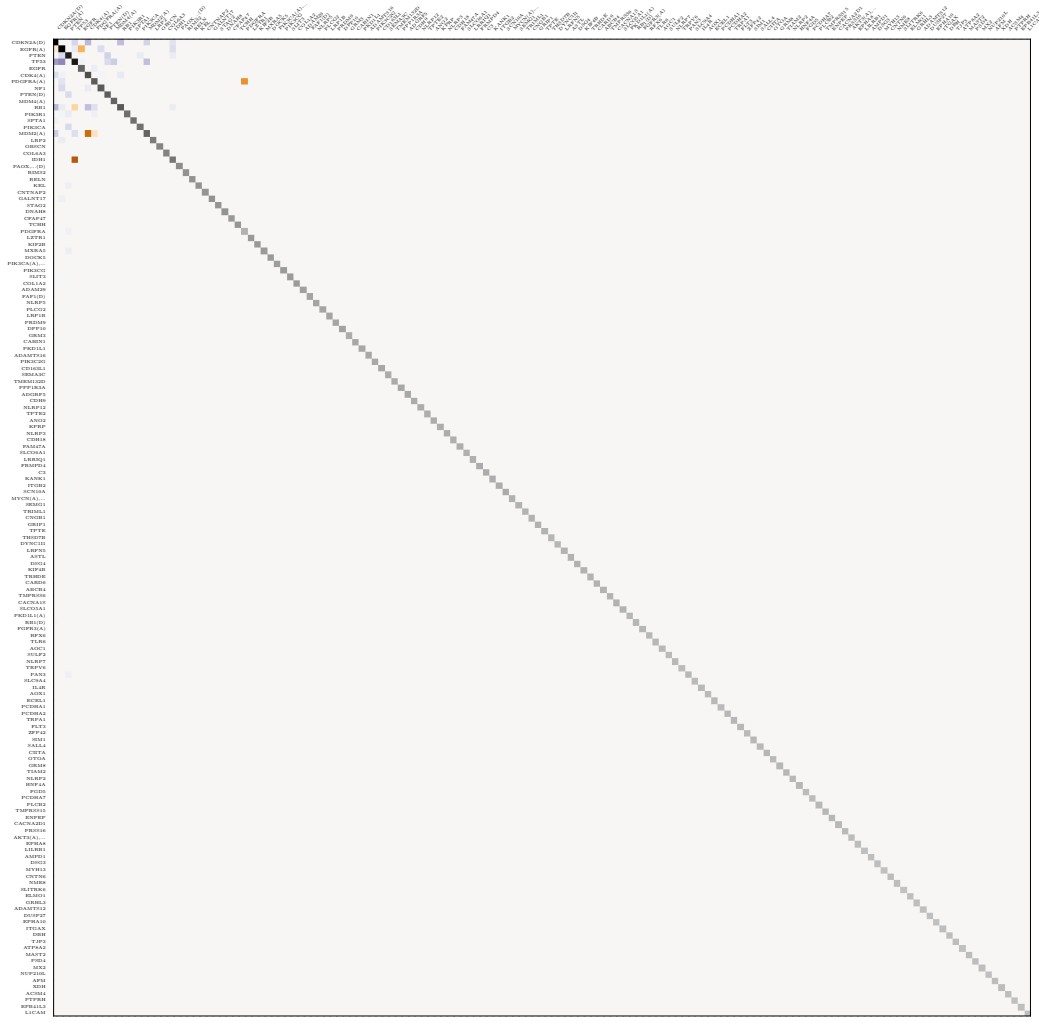

Figure 11: The learned parameter matrix $\Theta$ for the $n = 150$ most frequent genetic alterations in the TCGA glioblastoma data set discussed in Section 6. Note that the matrix consists of a smaller block of complex dependencies followed by a larger block of approximately independent items.