# OpenReview forum: "Scaling up Continuous-Time Markov Chains Helps Resolve Underspecification"
_NeurIPS.cc/2021/Conference — NeurIPS 2021 Poster_

### Official Review · Reviewer_ZrQu · 2021-07-13

**Rating:** 7
**Confidence:** 3

**Summary:**

This paper deals with learning a particular class of multi-component continuous time Markov chains (CTMC) when time stamps are unobserved in the data. In this class binary components change their state irreversibly, rates follow a pairwise interaction model and one component changes at a time. The authors show that modeling independent elements provides information about the posterior distribution of observation time and provide an efficient stochastic approximation to the likelihood’s gradient.

**Limitations And Societal Impact:**

I do not see a potential negative social impact that is specific to this work.

The paper lacks a discussion of limitations of the method. For example, the experimental section used a fixed value of lambda. This seems to have a major effect on the sparsity of the inferred interaction matrix. Is there a reasonable way to tune this hyperparameter as done in prediction tasks?


**Main Review:**

**Originality**:
Inferring temporal relations between sequences using independent mutations and a CTMC model is already a fundamental technique in phylogenetics. Therefore, I suggest mentioning the relation of the proposed approach to this field. Yet, as far as I know, the explicit quantification of the added value of independent components (Theorem 1) is novel.
The approximation technique is a novel combination of formulas for inferring the likelihood of partial sequences, stochastic gradient approximation and MCMC sampling.

**Quality**:
The added value of additional independent components is supported by a sound derivation in Equation 2. This added value is quantified in Theorem 1 which is proved in the appendix.
The approximation algorithm contains a series of well-established steps.
The experimental section gives a good illustration of actual gain of leveraging additional independent components and illustrate a very nice application of modeling cancer tumor progression.
While the title describes the method as a general CTMC, the actual model is a specific type that assume irreversible changes of components with pairwise influences on the rates. This subclass has an interesting important application and therefore it does not diminish the significance of this work. However, this specificity should be more clearly stated, at least in the abstract.

**Clarity**:
The paper is clearly written. Here are some suggestions to improve readability:
1.	‘Underspecified’ is a broad term and makes the abstract a bit vague. It may be good to state the sentence from the introduction -- ‘consist of unordered sets of item observed at unknown time point’ -- already in the abstract.
2.	The cancer example is very illustrative and provides a strong motivation. Furthermore, the paragraph in line 162 motivates the usage of independent mutations. It may be good to move it to the introduction, to make the motivation clearer at this stage.
Minor questions:
3.	Theorem 1: should V be replaced by V_{+}?
4.	Figure 1: do the different curves represent posteriors given different values of S_{+}?
5.	Where the N tumor samples taken from N distinct patients, or were multiple samples taken from some patients? This may influence the choice of modeling approach.

**Significance**:
This work may be an important step in enabling modelling of cancer tumor progression, and may have also applications in other domains in which systems accumulate irreversible changes over time with dependencies among some of the components.



**Time Spent Reviewing:**

6 hours

---

> ### Author Response · Authors · 2021-08-10
> **Author feedback**
>
> Thank you for your comments.
>
> We are happy to improve our manuscript based on your suggestions 1. and 2.
>
> 3. Yes, this is a typo. Thanks for pointing it out.
>
> 4. Yes, exactly.
>
> 5. Each tumor sample in our data was taken from a distinct patient.
>
> Regarding the choice of $\lambda$, in principle, one can use standard approaches like cross-validation to determine an appropriate value. The additional complication in models like this, is that the likelihood cannot be computed exactly for larger models, therefore we again have to resort to sampling-based approximations for this purpose. We are happy to add some discussion about this in the paper.

---

> > ### Comment · Reviewer_ZrQu · 2021-08-22
> > **Request for response about originality**
> >
> > Thank you for your feedback.
> >
> > Can you please add your response about the relation of your work to previous approached to infer temporal relations using independent component that are highly used in phylogeny analysis? (see 'Originality' section)
> > Specifically, what is the main novelty with respect to such works?

---

> > > ### Author Response · Authors · 2021-08-22
> > > **Response about originality**
> > >
> > > The two main points of novelty in our work are (a) the theoretical analysis of how the addition of independent items affects the inference of the CTMC parameters (Theorem 1), and (b) the formulation of the approximate likelihood maximization in section 4, which allows us to scale up the inference to hundreds of items. To our knowledge, neither of those are discussed in previous works on phylogenetics or other related disciplines that make use of CTMCs.
> > >
> > > We would also like to note that the problem formulation itself varies between applications in phylogenetics, cancer genomics, etc. For example, in our work (and some of the previous work we cite in our paper), we model an accumulation process of mutations. Other works in phylogenetics model a non-accumulating behavior; e.g., states may represent the bases ATCG, and the CTMC models the transitions between these states, which may include reverting to a previously-seen state.
> > >
> > > We are happy to add to our paper a comparison to previous work in phylogenetics.

---

### Official Review · Reviewer_HeAE · 2021-07-18

**Rating:** 7
**Confidence:** 3

**Summary:**

The authors tackle the problem of time evolution of discrete sets through the view of continuous-time markov chains. The authors provide a theoretical justification of why these problems in general underspecified in the setting of cross-sectional data. They go on to demonstrate that the underspecification can be overcome by use of additional items / features that can help determine time order by deriving a bound on the mean and variance of the observed time for these additional items. An approximate formulation for likelihood maximization is then presented without any assumptions of prior knowledge of the additional items or the structure of the additional items. The theoretical framework is then tested using simulation studies and demonstrate that accuracy of state sequence recovery can be significantly improved with use of the additional items. The framework is also evaluated using a real world cancer dataset from TCGA by ordering of pairs of mutation and copy number aberration events.

**Limitations And Societal Impact:**

The authors address the limitations of the behavior of the algorithm when assumption of independence of additional features is violated.

The work does not appear to lead to any potential negative societal impacts in my opinion.

**Main Review:**

This is a very interesting submission that demonstrates how underspecification in continuous-time markov chains can be resolved by use of additional independent features/ which might not directly relevant to the problem under consideration. The paper is well written and the use of a consistent example throughout the manuscript conveys the concepts clearly. My comments are largely related in practical applications:
1. The simulation results presented by the authors are convincing and a clear demonstration of the concept. However it is not clear to me how well the formulation would hold up under noisy data - perhaps larger markov chains where some observations might be missing subset of states would provide a practical view of the limitations
2. Any individual cancer type has been demonstrated to contain distinct sub types each with associated mutations and/or copy number alterations. For eg, the IDH mutation in GBM highlighted by the authors is observed in only a subset of individuals and is associated with better survival. Therefore a broad span of patient data will most likely include mutation /copy number aberration drivers that are distinct and mutually exclusive to each other. How is the learning process influenced in the presence of such confounding factors and how does it affect the interpretation of the results.



**Time Spent Reviewing:**

12

---

> ### Author Response · Authors · 2021-08-10
> **Author feedback**
>
> Thank you for your comments.
>
> While the focus of this paper is mainly on the scalability aspect, both of your points are important future considerations.
>
> 1. We would be interested in further investigating both theoretical and practical implications of different noise assumptions, possibly based on more detailed domain knowledge about the measurement and preprocessing procedures used to obtain the data. We would also be interested in extending the model to deal with missing data.
>
> 2. The effect of cancer subtypes on such analyses is worthy of a long discussion. Ideally, given enough data, one would hope that the different subtypes would be implicitly encoded in the inferred model. That is, different possible trajectories of the learned CTMC would potentially correspond to different sets and/or orders of genetic alterations that correspond to specific subtypes. However, it may also be worth thinking about ways to incorporate prior knowledge about specific subtypes into the learning process.

---

### Official Review · Reviewer_YnoU · 2021-07-18

**Rating:** 5
**Confidence:** 4

**Summary:**

The authors model the time evolution of discrete states (e.g. genetic mutations) using a continuous-time Markov chains. They show that the resulting learning task is generally underspecified given cross-sectional data. The authors suggest including additional independent items that can help determine time order, and resolve this underspecification.  The authors implement an approximate likelihood maximization method for learning continuous-time Markov chains, which shows good performance on high-dimensional data, is faster than competing methods, and shows promising results on synthetic and real cancer data.

**Limitations And Societal Impact:**

Limitations were not mentioned in the paper.

**Main Review:**

Originality: The work is a novel combination of well-known techniques. Related work needs to be adequetely cited

Quality: The submission is technically sound justified by theoretical analysis and experimental results. This is a complete piece of work and is built heavily using Schill 2019.

Clarity: The submission is clearly written.

Significance: The study and results are important in many fields. There is potential for others to use the ideas. The work provides a unique theoretical approach.

* Comments to authors

   * The problem setup section as well as Section 5.1 heavily lean on Schill 2019: https://academic.oup.com/bioinformatics/article/36/1/241/5524604
This fact is not clearly cited in the paper or Problem section.
It seems that the current paper builds on Schill 2019, while also uses the same cancer data for real-world experiments, with the value-add of theoretically proving the usage of additional data.

   * Regarding the additional unimportant data, how much and how complete should this be? The Markov chain would require that these additional data are present at (t-1), so that the chain can learn at t. What if additional data is missing at (t-1) or is available only intermittently? Would the equations in Section 4.2 still hold?

  * Is the maximum dim of \Theta_{full}, sum of dim of V and V_{+}? (Section 3.2). So in cases, when the dim of \Theta_{full} is < sum of dim of V and V_{+}, shouldn’t the off-diagonal information be taken into account as well?

  * From Theorem 1, Appendices C and G, it seems that \Theta_{full} has to maintain a block-diagonal structure. By adding additional unimportant data, across time points, there is a higher likelihood to see dependency across time points (i.e important signals in the off-diagonal elements). This is generally the case when identifying trajectories of cell differentiation in single-cell data across multiple (pseudo) time points and adding incomplete information for certain time points when available. It is a non-standard and underspecified problem, in which case, the dependency of additional terms is what helps connect time points. Based on this, aren’t the authors losing important information by enforcing a block-diagonal structure? How easy is it to incorporate information from off-diagonal elements into the CTMC?

**Time Spent Reviewing:**

4 hours

---

> ### Author Response · Authors · 2021-08-10
> **Author feedback**
>
> Thank you for your comments.
>
> * We have cited (Schill et al., '19) twice in the problem setup section (lines 72 and 94). We are happy to clarify this even further if needed. Our section 5.1 (synthetic experiments) does not rely at all on (Schill et al., '19).
>
> * As explained in the paper, the data is assumed to be cross-sectional. That is, a data sample consists of a set of observed mutations at some (unknown) continuous time $t$. Time is not discrete, and there is no information about data at $(t-1)$ available.
>
> * While our theoretical results rely on a block-diagonal assumption, in practice this is not strictly enforced, but rather implicitly encouraged by the regularization term (see eq. (3)). The purpose of the experiments in Appendix C is exactly to test what happens when there are dependencies (positive or negative) between the "unimportant" elements. The results show that the model is fairly robust to these dependencies. Similarly, in the inferred matrix shown in appendix G, we do not explicitly enforce a block-diagonal structure, but rather this structure emerges from the data itself and also contains some smaller interactions involving "unimportant" elements.

---

### Official Review · Reviewer_SqMU · 2021-08-01

**Rating:** 6
**Confidence:** 4

**Summary:**

This paper describes a continuous-time Markov Chain method for modeling the time evolution of events. By including independent items the authors are able to increase the limit on the number of items modeled. This research is particularly relevant as larger single-cell sequencing data sets are being collected.

**Limitations And Societal Impact:**

  The authors only compared to Schill (2019). There are many tumor evolution method available in the bioinformatics literature and without a more comprehensive comparison, the utility of this approach may not be clear and adoption of the approach may be limited.

**Main Review:**

  The underlying biomedical problem is well-motivated and the authors briefly mention other possible applications. I think they could have provided more clarity on the specific problem --- the use of the term "item" is confusing in paragraph 2. The reader wonders if an "item" is a cell or a genetic locus, though this is clarified later in the paper.

  Going from cross-section data to longitudinal inferences requires assumptions about the distribution of the samples. It would be helpful to be more clear about the assumptions that enable the underlying inferential goal of this work.

  The "warm-up" example clearly illustrates an important point. But the title "warm-up" seems pedantic. Perhaps a more descriptive title such as "A counter-example for inferring time order" would help the reader to understand the motivation of the example. The discussion in section 3.3 is helpful in that it links back to the counterexample.

  There are some standards for genetic nomenclature and it does not seem the case that the authors have used that nomenclature here. This leads to some confusion as to what exactly is meant by "amplification" in figure 3. Is this an increase in copy number or increase in transcription or translation due to an epigenetic change or something else. It would be helpful to clearly state what the data can and can't inform about the mechanisms.

  The meaning of the error bars in figure 3 is not clearly stated.

  The development of the method is predicated on a set of "interesting items" $V$ and a set of "independent items" that are used to help inform the clock (lines 1-168). Then, in the paragraph on lines 169-174, the authors state, "we will make no distinction between 'items of interest' and independent items". It is not clear, theoretically, what motivates or justifies this shift in perspective. Prior to this change, I was left considering how one might select "interesting" and "independent" genetic loci, but it seems that has been rendered unnecessary. It would be helpful for the authors to clarify what motivates and theoretically justifies the elimination of the need to specify $V_+$ - the "independent" set of items.

  The description of the experiments makes it impossible to reproduce the results of the paper. On line 280, the authors state that they evaluate on a data set of tumor samples from glioblastoma from TCGA, but TCGA contains expression, mutation, and methylation data. It is not clear what data the authors used and the url to the entire project does not enable reproduction of the analysis. Perhaps a more detailed description with code can be provided in supplementary information.

**Time Spent Reviewing:**

2

---

> ### Author Response · Authors · 2021-08-10
> **Author feedback**
>
> Thank you for your comments.
>
> * We are happy to improve on the exposition clarity and nomenclature based on your suggestions.
>
> * All error bars represent +/- 2 standard errors of the mean.
>
> * The existing data makes no distinction between "interesting" and "independent" alterations. While it would be possible to use some prior knowledge to split the alterations into two groups, it seems more practical to let the model figure this out. However, we do know that a large amount of the alterations are (approximately) independent, even without knowing in advance exactly which ones have this property. As a result, on a theoretical level we wanted to investigate the effect that this particular structure of the data has on model inference, specifically with regards to estimating the observation time, as well as model identifiability. In practice, the benefit of these independent alterations is gained even without pre-specifying them.
>
> * As mentioned in the paper, we use point mutation and copy number alteration (amplification, deep deletion) data from TCGA, obtained via cBioPortal. We use no expression or methylation data in our experiments. We have included all code and data that we used for our experiments in the supplementary material attached to our submission. This should make it straightforward to reproduce all synthetic and real data results presented in the paper.

---

### Decision · Program_Chairs · 2021-09-27

**Decision:**

Accept (Poster)

**Comment:**

This paper describes a method for modeling the time-series changes in a discrete set of items. The particular application the authors have in mind is phylogenetic inference where the items are genetic mutations. The method involves reframing and using continuous-time Markov chains. While moving to the continuous domain helps address the combinatorial nature of the problem, it introduces another problem - underspecification - which results in poor generalization, sensitivity, and false positive correlations. The authors propose a solution based on prior work (Schill et. al 2019) that employs so-called "unimportant items" to help recover the time order.

The reviewers noted two specific aspects of significance of the work. First, the paper provides bounds on the expected amount of information gain from including these "unimportant items". Second, the paper addresses an important problem in bioinformatics. Inferring the tumor evolution process is a well-studies problem in genomics and has been studied extensively. In clinical applications, the tumor evolution may be of limited concern because the physician is primarily interested in treating the whole tumor as it presents (including any heterogeneity). But, in fundamental research, understanding the tumor evolution can inform our basic understanding of the processes that lead to tumor growth and metastatic processes and therefore phylogenetic inference is a significant problem to address. It should be noted that for this method to be used in practice, a distinction must be made between "interesting" and "uninteresting" items. This may be done by the model or with prior knowledge. The authors have constructed their algorithm to allow the model to make this distinction which may improve the generality for problems outside of phylogenetic inference.

I'm in agreement with the reviewers after considering the author response and discussion that this paper should be recommended for acceptance. The theoretical contributions may prove useful for a broad range of similar problems and the practical utility on a data set relevant for practical phylogenetic inference is demonstrated.